*Report*

# RNase III cleavage sites spread across splice junctions enforce sequential snoRNA processing

Valérie Migeot[1], Yves Mary[1], Etienne Fafard-Couture[2], Pierre Lombard[1], François Bachand [2], Michelle S Scott [2] & Carlo Yague-Sanz [1✉]

## Abstract

Small nucleolar RNAs (snoRNAs) are a class of eukaryotic non-coding RNA molecules whose precursor transcripts are capped and polyadenylated. However, these end modifications are detrimental to snoRNA function and must be removed, a process typically involving excision from introns and/or endonucleolytic cleavage. For RNA precursors that host multiple snoRNAs, the sequence of maturation events is potentially important, but not well understood. Here, we report a new mode of maturation concerning snoRNA pairs that are co-hosted in the intron and the adjacent 3′ exon of a precursor transcript. For a snoRNA pair with this arrangement in *Schizosaccharomyces pombe*, we found that the sequence surrounding an exon–exon junction within their precursor transcript folds into a hairpin after splicing of the intron. This hairpin recruits the RNase III ortholog Pac1, which participates in the maturation of the downstream snoRNA by cleaving the precursor. Our findings suggest that conditional RNase III cleavage signals hidden in an exon–exon junction evolved to enforce sequential snoRNA processing. Sequence analysis suggests that this mechanism is conserved in animals and plants.

**Keywords** Drosha; Maturation; Pac1; RNAse; snoRNA
**Subject Category** RNA Biology

## Introduction

Small nucleolar RNAs (snoRNAs) constitute a class of eukaryotic non-coding RNA that guide chemical modifications on other RNA molecules. Most snoRNAs can be classified into one of two major snoRNA families based on distinct structural features, functions, and conserved sequence motifs (or boxes). The C/D box snoRNAs fold into a closed loop structure and associate with methyltransferase proteins to deposit 2′-O-methylation on target RNAs. In contrast, H/ACA box snoRNAs adopt a double hairpin structure and interact with pseudouridine synthase proteins to guide pseudouridination. Members of both snoRNA families specifically recognize their targets through base-pairing interactions between guide sequences in the snoRNA and complementary sequences in the substrate RNA (Dupuis-Sandoval et al, 2015; Huang et al, 2022). One of the major targets for snoRNA-directed modification is ribosomal RNA (rRNA), which aligns with the preferential nucleolar localization of snoRNAs.

SnoRNA maturation and biogenesis in eukaryotes rely on complex mechanisms that are often perceived as unusual compared to other gene classes. Being transcribed by RNA polymerase II (RNAPII), snoRNA precursor transcripts (pre-snoRNAs) undergo co-transcriptional capping and polyadenylation, similarly to messenger RNAs (mRNAs). The 7-methylguanosine ($m^7G$) cap and poly(A) tail are essential modifications that protect mRNA from exonucleolytic degradation and facilitate their export and translation in the cytoplasm (Gallie, 1991). In contrast, mature snoRNAs are intrinsically stable due to their structure and the association of partner proteins, and do not require end modifications—which can even be detrimental to snoRNA function. For instance, $m^7G$-capped snoRNA in yeast have been shown to mislocalize to the cytoplasm, where they are targeted for degradation (Grzechnik et al, 2018). Therefore, the $m^7G$ cap and poly(A) tail need to be removed or altered from pre-snoRNAs, relying on distinct mechanisms depending on snoRNA genomic organization, reviewed in (Kufel and Grzechnik, 2019) and briefly summarized hereafter.

In mammals, snoRNAs are often encoded in introns of host genes. For instance, 61% of all annotated snoRNAs in human are intronic (Fafard-Couture et al, 2024; Bergeron et al, 2023). Following splicing, intronic snoRNAs can be excised from the lariat either by endonucleolytic cleavage or by the combined action of the debranching enzyme and exonucleases, resulting in cap- and poly(A)-less linear transcripts. Remarkably, intronic snoRNAs are often hosted in genes coding for proteins involved in ribosome synthesis, suggesting that snoRNA and ribosome production are coordinated. However, host genes are not exclusively protein coding: a significant fraction of intronic snoRNA (23% in human) (Bergeron et al, 2023; Fafard-Couture et al, 2024) are hosted in long non-coding RNAs (lncRNA) genes, most of which have no apparent function beyond serving as snoRNA precursors.

Non-intronic snoRNA, which account for 90 and 81% of all snoRNAs in budding yeast and fission yeast, respectively (Fafard-Couture et al, 2024), can be organized either as polycistrons or as individual genes. For individually encoded snoRNAs, 3′ end processing differs among species. In fission yeast, it is mediated by the mRNA

[1]URPHYM-GEMO, NARILIS, University of Namur, Namur, Belgium. [2]RNA Group, Department of Biochemistry & Functional Genomics, Université de Sherbrooke, Sherbrooke, QC, Canada. ✉E-mail: carlo.yaguesanz@unamur.be

cleavage and polyadenylation machinery (Larochelle et al, 2018), with subsequent deadenylation by the nuclear exosome guided by the nuclear poly(A)-binding protein Pab2 (Lemay et al, 2010). 5′-end processing differs between snoRNA families with most individual C/D box pre-snoRNAs being transcribed with a capped 5′ leading sequence which is subsequently removed by endonucleolytic cleavage and exonucleolytic trimming (Chanfreau et al, 1998a; Lee et al, 2003). Conversely, for pre-snoRNA transcribed without a leading sequence (including most H/ACA snoRNA in yeast), the m⁷G cap of the pre-snoRNA is converted into a trimethylguanosine ($m^{2,7,7}$G or TMG) cap (Terns and Dahlberg, 1994).

SnoRNA genes can also be organized into operons under the control of a single promoter, where individual snoRNAs are typically matured from polycistronic transcripts through endonucleolytic cleavage between the cistrons. In budding yeast, this process is mediated by the RNase III family protein Rnt1 (Chanfreau et al, 1998a; Ghazal et al, 2005). In other species, including fission yeast and human, the actors mediating the 5′ and 3′ maturation of non-intronic snoRNA co-hosted in the same precursor transcripts remain largely unknown. In these cases, the sequence of maturation events may be particularly significant, as precursor cleavage exposes snoRNAs and surrounding sequences to exonucleolytic degradation. Whether and how the order of polycistronic snoRNA processing is controlled is also unknown.

In this study, we explored these questions using the model eukaryote *Schizosaccharomyces pombe*. In *S. pombe*, the RNase III homolog Pac1 is an endoribonuclease involved in RNA polymerase II termination and ncRNA biogenesis, including the 5′ maturation of several C/D box snoRNAs (Yague-Sanz et al, 2021). Mechanistically, Pac1 targets double-stranded RNA (dsRNA) hairpins with a minimum stem length of 20 base pairs, tolerating small bulges and varied loop structures (Rotondo et al, 1997; Ivakine et al, 2003), and cleaves them in two staggered cuts. This contrasts with its homolog Rnt1 in budding yeast that specifically targets for cleavage RNA hairpins with apical NGNN tetraloops (Lamontagne and Elela, 2004).

Here, we report a new role for Pac1 in the maturation of a snoRNA pair co-hosted within the intron and adjacent 3′ exon of a precursor transcript. Remarkably, Pac1 activity at this locus depends on a dsRNA hairpin that folds across the exon–exon junction after splicing of the intron. Our findings suggest that this exon–exon junction serves as a conditional degradation signal that ensures sequential snoRNA processing, a mechanism that may be conserved across eukaryotes, including animals and plants.

## Results and discussion

### snoU14 and mamRNA are transcribed from a polycistronic precursor stabilized by Pac1 inactivation

Based on high-throughput genomic and transcriptomic data, we previously identified Pac1 cleavage signatures on at least five C/D box snoRNA in *S. pombe*, including snoU14, for which Pac1 inactivation lead to the accumulation of 5′-extended precursor transcripts (Yague-Sanz et al, 2021). Accordingly, we proposed that Pac1 is involved in snoRNA maturation through 5′-end processing (Yague-Sanz et al, 2021). However, the discovery by the Rougemaille group of a previously unannotated ncRNA named *mamRNA* (for *Mmi1* and *Mei2-associated RNA*) directly upstream of snoU14 (Andric et al, 2021) suggests that Pac1's role at this locus

may be more complex than previously thought. Moreover, recent work from the same research group—published while this manuscript was in review—describes that mamRNA and snoU14 are expressed from a common precursor transcript (Leroy et al, 2025), a conclusion we also reached independently when evaluating the contribution of Pac1 to mamRNA/snoU14 expression. This common precursor is also host to another C/D box snoRNA, snR107, located in its unique intron (Leroy et al, 2025) (Fig. 1A).

To assess Pac1's role in modulating mamRNA expression, we first performed northern blot experiments using radiolabeled probes complementary to the mamRNA (*C*), snoU14 (*B*) and the intergenic region between the two genes (*A*) under conditions where the activity of Pac1 is compromised. Specifically, we used either the Pac1-AA (anchor-away) strain, which enables the conditional nuclear exclusion of Pac1, or the Pac1-ts (thermosensitive) strain, which carries a hypomorphic Pac1 allele that impairs Pac1 activity even at growth-permissive temperature (Yague-Sanz et al, 2021). As expected, the *A* and *B* probes recapitulated our previous observation that a long 5′-extended precursor containing the mature snoU14 sequence accumulates in conditions of Pac1 deficiency (Fig. 1B, *A-B* probes). The mamRNA probe revealed at least four distinct transcripts (Fig. 1B, *C* probe). Among these, the two shortest ones were the most abundant within the control strains (when Pac1 is active), with length matching the annotated mamRNA *short* and *long* isoforms (Andric et al, 2021; Lock et al, 2017). The longest mamRNA-containing transcript slightly accumulated in the Pac1-ts strain only, while the second longest transcript consistently accumulated in both Pac1-deficient strains. Since these isoforms migrated at the same size as the 5′-extended snoU14 precursors, we tentatively named them *pre-mamRNA/snoU14*. Long read RNA sequencing validated the existence these isoforms containing both the mamRNA and snoU14 sequences (Fig. EV1A), as also recently reported (Leroy et al, 2025). Further suggesting that they are transcribed together, previously published Cap Analysis of Gene Expression (CAGE) data (Thodberg et al, 2019) highlighted a single transcription start site for the entire locus, located directly upstream from the mamRNA gene (Fig. 1D). Collectively, these results confirm that snoU14 and mamRNA are transcribed from a single precursor (Leroy et al, 2025) and show that the abundance of a form of this precursor depends on Pac1 activity.

We then corroborated our northern blot results with splice-specific RT-qPCR experiments, confirming that precursor transcripts from the locus were upregulated in Pac1-deficient strains (Fig. 1C). This upregulation was significant with the amplicon spanning the exon–exon junction within the mamRNA, but not for the amplicons spanning the exon–intron junction or the intronic region, suggesting that the isoform consistently affected by Pac1 activity was spliced. This was further confirmed by short-read RNA sequencing that showed a specific accumulation of spliced reads spanning the exon–exon junction (Fig. 1D,E).

Although mature levels of snoU14 and snR107 remained unaffected by Pac1 inactivation in the Pac1-AA strain, they increased in the Pac1-ts strain (Figs. 1B and EV1B). Similarly, unspliced precursor levels were also slightly increased in the Pac1-ts strain (Fig. 1B,C). These strain-specific accumulations might be indirectly caused by long-term adaptations of the Pac1-ts strain, where Pac1 is deficient even at growth-permissive temperature. Nevertheless, we concluded from these experiments that the loss of Pac1 activity

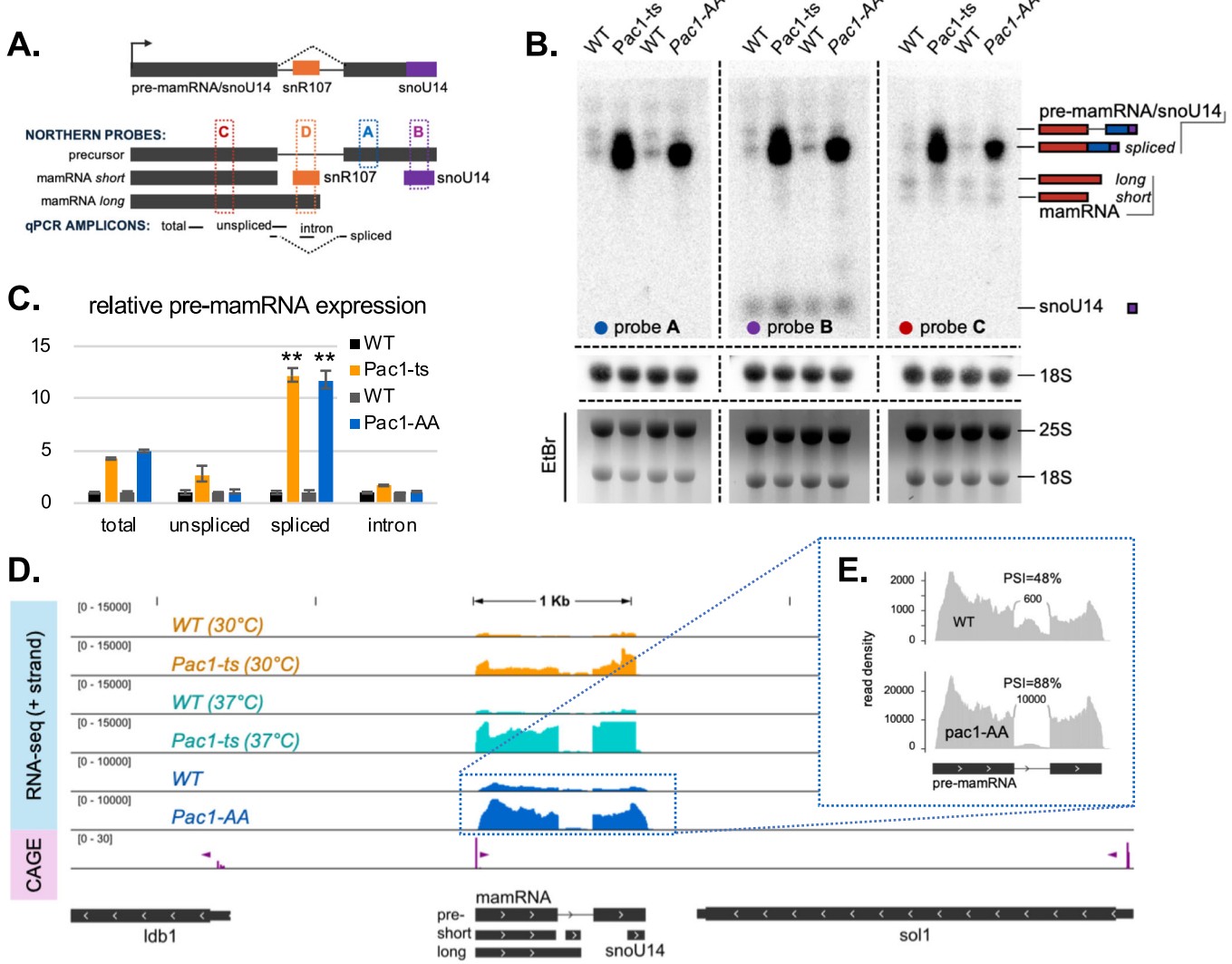

**Figure 1.  A common precursor for mamRNA and snoU14 depends on Pac1 activity.**

(A) mamRNA-snoU14 locus representation. The arrow indicates the unique transcription start site for the locus. Intron is represented as a straight line, while exons are represented as rectangles. Co-hosted snoRNAs snR107 and snoU14 are represented in orange and purple, respectively. Northern blot probes and qPCR amplicons are positioned relative to the indicated transcript isoforms originating from the locus. (B) Representative northern blot analysis of transcripts originating from the mamRNA/snoU14 locus under conditions where the activity of Pac1 is compromised. (C) RT-qPCR analysis (n = 3) of transcript isoforms spanning the mamRNA locus. Statistical significance of the differences in spliced isoform level in the Pac1 mutants compared to their respective control strains is indicated (**: Student's t test P value = 0.0044 and 0.0024, respectively). Error bars represent the standard deviation of the mean. (D) Genome browser snapshot of the normalized read density over the mamRNA/snoU14 locus for RNA-seq experiments (+ strand only) and CAGE-seq (+ and − strands). For the CAGE-seq track, the direction of transcriptional firing is indicated with purple arrowheads. Gene annotation is shown in dark gray. (E) Sashimi plot depicting the junctions reads as arcs over the read density profiles at the mamRNA locus for the Pac1-AA and matching WT RNA-seq experiments. The number of junction reads (rounded to the nearest 100) and percent spliced-in values (PSI) are indicated. EtBr Ethidium Bromide, ts thermosensitive, AA anchor-away, WT wild-type. Source data are available online for this figure.

increased the expression levels of mamRNA/snoU14 precursors, an effect that is only consistent across conditions for the spliced isoform.

## A stemloop structure at the pre-mamRNA exon–exon junction directs Pac1 cleavage

We next wanted to identify *cis*-acting elements in the pre-mamRNA/snoU14 transcript responsible for Pac1 recruitment and/or activity. The sequence directly upstream of snoU14 has previously been predicted to fold in a stemloop which could serve as a Pac1 substrate;

however, this has not been experimentally tested (Yague-Sanz et al, 2021). To challenge this prediction, we introduced eight point mutations expected to dramatically disrupt the predicted structure and named the resulting mutant *snoU14-SD* (S*tem*D*ead*) (Fig. 2A). In this strain, the mature snoU14 levels were significantly decreased by about 34% (Fig. 2B, *B* probe; quantified in Fig. EV2A), suggesting that the mutated sequence might play a role in snoU14 expression. However, the snoU14-SD mutant did not phenocopy Pac1 deficiency: the expression of the spliced pre-mamRNA/snoU14 isoform remained at wild-type levels, as measured by both northern

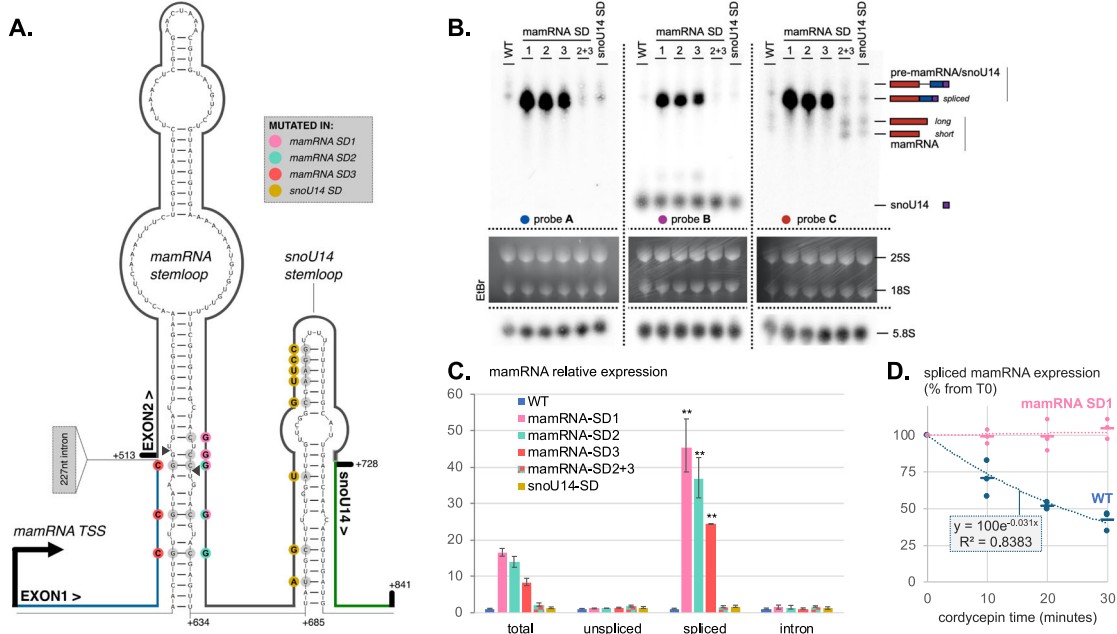

**Figure 2. A stemloop spanning the exon–exon junction directs Pac1 cleavage at the mamRNA.**

(A) Predicted secondary structures of the spliced pre-mamRNA/snoU14 transcript. Nucleotide positions are numbered relatively to the mamRNA transcription start site (TSS). Mutated bases are color-coded according to the StemDead (SD) mutants in which they were altered. Putative Pac1 cleavage sites, inferred from degradome-seq, are indicated by arrowheads. (B) Northern blot analysis of transcripts originating from the mamRNA/snoU14 locus. Probe positions are indicated in Fig. 1A. EtBr ethidium bromide. (C) RT-qPCR analysis ($n = 3$) of transcript isoforms spanning the mamRNA locus. The qPCR amplicons are indicated in Fig. 1A. Statistical significance of differences in spliced isoform levels between SD mutants and their control strain (WT) is indicated (**: Student's $t$ test pvalue = 0.0026, 0.0039 and 0.0016, respectively). Error bars represent the standard deviation of the mean. (D) Decay rate of the spliced pre-mamRNA/snoU14 transcript, measured by RT-qPCR after transcriptional inhibition with 0.6 mM cordycepin ($n = 3$). Error bars represent the standard deviation of the mean. Source data are available online for this figure.

blot and splice-specific RT-qPCR (Fig. 2B,C). From these observations, we conclude that the predicted stemloop proximal to snoU14 was not responsible for Pac1 activity on the precursor transcript.

We then scanned both the unspliced and spliced pre-mamRNA/snoU14 sequences for other RNA structures that could potentially serve as Pac1 substrate. Strikingly, a very stable stemloop ($\Delta G = -35$ Kcal/mol) with symmetric bulges was predicted to fold across the exon–exon junction ($-15$ of exon 1 to $+121$ of exon 2) of the spliced precursor (Fig. 2A). Suggesting that this stemloop could constitute a genuine Pac1 substrate, degradome-seq experiments (Zhang and Pelechano, 2021) revealed that the 5′-ends of two cleavage intermediates (Fig. EV2C) map on its structure and show the typical 2nt-overhang signature of RNase III cleavage (Fig. 2A, arrowheads). Importantly, this stemloop was not predicted in the unspliced isoform, which instead formed a branched structure due to the disruptive presence of the intron (Fig. EV2B).

To verify that the identified stemloop was responsible for Pac1 activity at the locus, we introduced three sets of point mutations in the endogenous mamRNA gene, generating the *mamRNA-SD1*, *-SD2*, and *-SD3* mutants (Fig. 2A). The disruptive mutations of the *mamRNA-SD1* and SD2 mutant are located in the exon 2, near the 3′ end of the predicted structure. Conversely, the disruptive mutations in the mamRNA-SD3 mutant are located in the exon 1, near the 5′ end of the predicted structure. The SD2 and SD3 mutations were designed so that their combination restores base-pairing, reforming the structure that the individual mutations disrupted, and therefore allowing us to discriminate between sequence- and structure-dependent effects.

In all individual mamRNA-SD mutants (mamRNA-SD1, SD2, and SD3), the steady-state levels of the spliced pre-mamRNA/snoU14 isoform were dramatically increased (Fig. 2B,C) while the mature mamRNA forms were less abundant (Fig. 2B, C probe). Strikingly, these effects were not observed in the mamRNA-SD2+3 mutant where the SD2 and SD3 complementary mutations reform a stable structure. These results strongly suggest that the RNA secondary structure spanning the exon–exon junction of mamRNA is responsible for the Pac1-mediated repression of the spliced pre-mamRNA/snoU14 isoform. Confirming this hypothesis, we found that mutations disrupting the secondary structure were epistatic to Pac1 loss of function, as Pac1 exclusion from the nucleus in the mamRNA-SD1 background did not further increase the levels of the spliced pre-mamRNA/snoU14 isoform (Fig. EV2D).

Arguably, the most straightforward explanation for how Pac1 represses the *pre-mamRNA/snoU14* isoform is that its endonucleolytic activity destabilizes the transcript. To test this hypothesis, we measured the decay rate of mamRNA isoforms in a time-course experiment following transcription inhibition with the chain-terminating adenosine analog cordycepin (see "Methods" section for details about inhibitor selection). In wild-type cells, cordycepin treatment caused a rapid decrease in the relative abundance of the spliced mamRNA/snoU14 precursor (estimated half-life assuming a complete transcriptional inhibition = 20 min). However, in the mamRNA-SD1 mutant, the isoform was drastically stabilized, with no detectable decay over the course of the experiment (Fig. 2D). From these results, we concluded that the stemloop structure over

the splice junction destabilizes the spliced mamRNA-snoU14 precursor in a Pac1-dependent manner.

## snoU14 can mature independently from Pac1 activity

The degradation signal localized at the mamRNA splice junction raises the question of its role in snoRNA maturation. We initially hypothesized that Pac1 cleavage at this locus was necessary to release snoU14 from the mamRNA precursor. Supporting this idea, northern blot experiments showed that snoU14 is sequestered in the longer precursor transcript when Pac1 activity is compromised in Pac1-deficient strains or mamRNA-SD mutants (Figs. 1B and 2B, B probe). However, mature snoU14 levels were not reduced, suggesting that alternative pathways are able to mature snoU14 from its precursor, potentially with slower kinetics.

SnoU14 is a conserved and essential C/D box snoRNA required for the maturation of the 18S rRNA from the polycistronic 35S rRNA precursor. Accordingly, deficiencies in snoU14 expression have been reported to severely affect rRNA processing, 18S levels, and cellular growth in yeast (Samarsky et al, 1996; Zagorski et al, 1988). Here, northern blot analysis on rRNA precursors showed that reduced snoU14 expression in the snoU14-SD strain (Fig. 2B) correlated with a modest (~20%) but consistant increase in 35S and 33/32S rRNA precursor (Fig. EVs 3A,B). In contrast, no perceptible differences in mature or precursor 18S rRNA levels could be observed in the mamRNA-SD mutants (Fig. EV3A,B), indicating that snoU14 matured by putative Pac1-independent pathway remain fully functional. Consistently, all the mamRNA-SD strains grew normally in rich medium (Fig. EV3C). From these experiments, we conclude that Pac1 cleavage on the pre-mamRNA/snoU14 transcript is not strictly required for snoU14 expression or function.

## Pac1 cleavage across the splice junction coordinates the sequence of snoRNA maturation events

Disjoint secondary RNA structure spanning splice junctions constitute an unique arrangement that forces RNase III cleavage to occur only after intron splicing, a condition that may be required for proper expression of the co-hosted intronic snoRNA. To test this hypothesis, we re-engineered the mamRNA/snoU14 locus by inserting a copy of the last 15 nucleotides of the exon1, corresponding to the 5′ end of the stemloop, at the beginning of the exon2 (Fig. 3A). In the resulting strain, mamRNA-CS$_1$ (ConstitutiveStemloop 1), folding of the hairpin should be uncoupled from intron splicing, potentially leading to faster or unconditional Pac1 cleavage. An alternative version of the mutant (mamRNA-CS$_2$), preserves the original sequence immediately downstream of the 3′ splice site and immediately upstream of the 5′end of the hairpin (Fig. 3A). Strains carrying either the mamRNA-CS$_1$ or -CS$_2$ mutations grew normally in rich medium (Fig. EV3D). Importantly, these mutations suppressed the increased levels of spliced precursor in the SD$_3$ mutant (Fig. 3B, A-B probes; Fig. 3C), demonstrating that the re-engineered "constitutive" stemloops act as functional cleavage substrates for Pac1.

Northern blot analysis of the CS mutants revealed that a small fraction of intronic snR107 remained trapped in a longer RNA species (Fig. 3B, D probe) whose 3′ end terminates two nucleotides downstream of snR107, as shown by 3′ RACE (Fig. EV3E). This transcript therefore resembles the previously annotated mamRNA

long isoform (Fig. 3D) (Lock et al, 2017; Andric et al, 2021), though it is 15 nucleotides shorter, and contributes to the apparent increase in its expression in CS mutants (Fig. 3B, C probes). These findings suggest that unconditional Pac1 cleavage can cause aberrant snR107 processing, although not to an extent sufficient to reduce mature snR107 level. A possible explanation for this observation is that Pac1 cleavage stochastically occurring before splicing on the mamRNA/snoU14 precursor could lead to 3′->5′ exonucleolytic trimming of the 3′ splice site, antagonizing splicing and subsequent maturation of snR107 (Fig. 4A). Consistent with this model, qPCR analysis showed significantly increased levels of unspliced isoforms in the mamRNA-CS strains (Fig. 3C).

## Conservation of secondary structures spanning splice junctions in mixed snoRNA clusters

Our observations suggest that a cleavage signal spread across splice junctions enforces sequential processing of snoRNA pairs co-hosted in the intron and adjacent exon of the same precursor transcript (Fig. 4A). We next wondered whether a similar mechanism might also exist in other eukaryotes. In related Schizosaccharomyces species, the peculiar organization of the snR107-snoU14 locus is conserved, with disjoint secondary structures spanning the splice junctions of the intron that hosts snR107 (Fig. EV4A,B). However, the distance between conserved elements (splice-sites, snoRNAs, and structures) varies, indicating that surrounding sequences are less conserved compared to these functionally constrained features. Structure and sequence alignment of the predicted secondary RNA structures revealed that the apical regions are less conserved than the base of the stems, particularly near the splice and cleavage sites. In these regions, we observed compensatory or base-pair-neutral mutations (e.g., U-to-C or C-to-U, which both pair with G), suggesting structural conservation (Fig. EV4B,C). This result is consistent with Pac1 substrate specificity that is very tolerant to variation in the loop (Rotondo et al, 1997), but may have kept sequences close to cleavage sites under selective pressure.

In more divergent organisms, conservation of the snR107-snoU14 locus (with snR107 in a intron) was not evident. This low level of conservation was not unexpected given snoRNAs high genetic mobility (Hoeppner and Poole, 2012). Regardless, we reasoned that our model (Fig. 4A) could apply to other snoRNA clusters organized similarly. Therefore we mined the Ensembl genome annotation (Harrison et al, 2024) for snoRNA precursors that are host to at least one intronic snoRNA and one non-intronic snoRNA, focusing on seven model species known for their high-quality annotation: S. cerevisiae, Caenorhabditis elegans, Drosophila melanogaster, Mus musculus, Rattus norvegicus, Arabidopsis thaliana, and Homo sapiens.

In total, we identified twelve of these mixed (intronic/non-intronic) snoRNA clusters. None were present in the intron-poor S. cerevisiae genome, whereas between one and three were found in the other species considered (Dataset EV1). These findings suggest that, although relatively rare, mixed snoRNA clusters exist in species beyond S. pombe. Among the identified clusters, the majority (ten out of twelve) are organized such that the last hosted snoRNA is located in an exon of the precursor transcript, preceded by one or more intronic snoRNAs (Fig. EV4D). Of these ten clusters, at least seven harbored predictable RNA stemloop

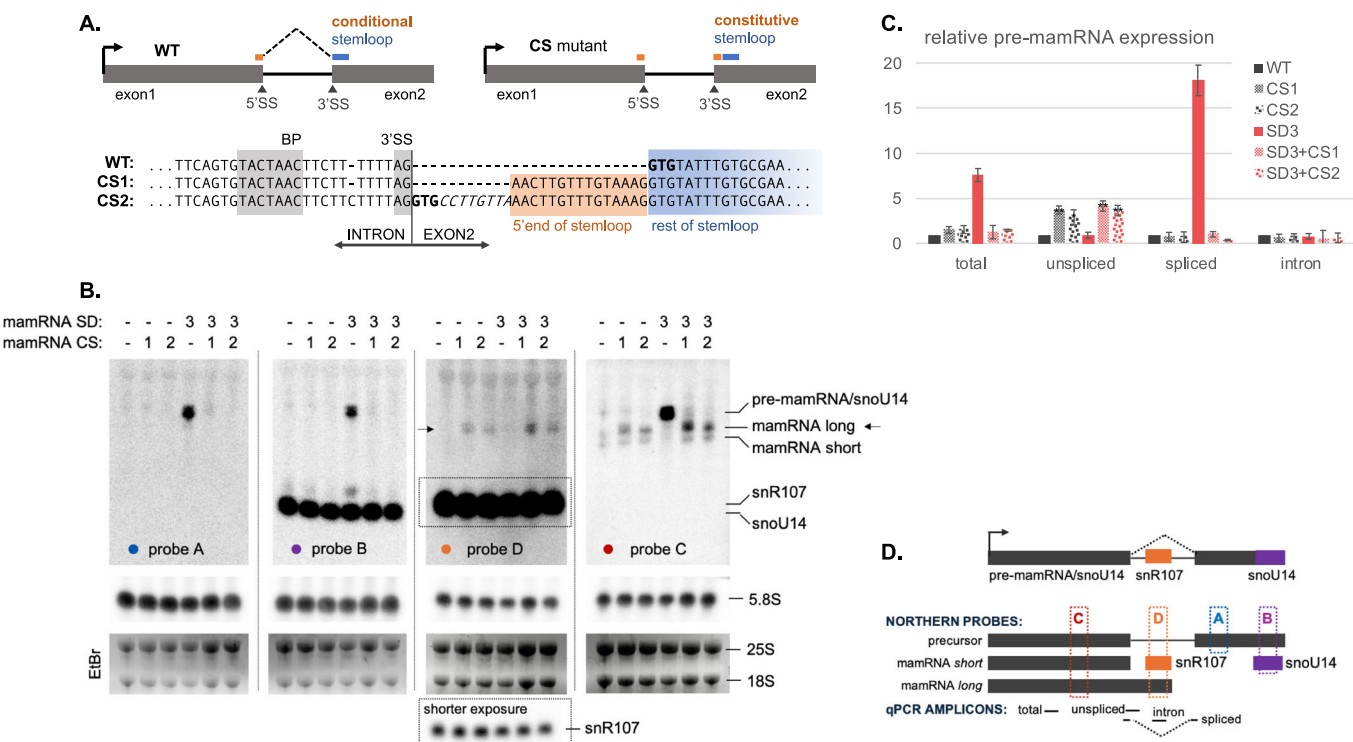

**Figure 3. Conditional Pac1 cleavage ensures sequential snoRNA processing.**

(A) Top: schematic representation of the mamRNA Constitutive Stem (CS) mutations. Bottom: mutated sequences of the mamRNA-CS mutants. For the mamRNA-CS$_2$ mutant, the nucleotides in bold and italic correspond to the original WT sequence immediately downstream the 3′ splice site and immediately upstream the 5′end of the stemloop, respectively. SS splice site, BP branch point. (B) Northern blot analysis of transcripts originating from the mamRNA/snoU14 locus. Probe positions are indicated on the right. Arrows indicate the snR107-containing long mamRNA isoform. EtBr: ethidium bromide. (C) RT-qPCR analysis ($n = 3$) of transcript isoforms spanning the mamRNA locus in the indicated strains. Statistical significance of the differences in unspliced isoform level in the CS mutants compared to their control strains is indicated (*: Student's $t$ test $P = 0.028$; **$P = 0.0024$). Error bars represent the standard deviation of the mean. (D) mamRNA-snoU14 locus representation as described in Fig. 1A. Source data are available online for this figure.

structures folding from the sequences spanning the last two exons of their spliced host precursor (Fig. EV4E).

To assess whether these secondary structures could also function as processing signals, we focused on the *SNHG25* snoRNA host gene in human. Re-analysis of long read nascent RNA sequencing (nano-COP) (Drexler et al, 2020) and CAGE-seq (Jakobsen et al, 2024) datasets indicated both the intronic (SNORD104) and non-intronic (SNORA50C) snoRNAs encoded in the *SNHG25* genes are present in the same nascent precursor transcripts transcribed from a single major transcription start site, validating the *SNHG25* gene structure (Figs. EV4D and 4B). Moreover, degradome-seq experiments in human cell lines (Cass et al, 2016) indicated that the predicted splicing-dependent stemloop in the *SNHG25* RNA is subject to endonucleolytic cleavage (Fig. 4B). The cleavage sites are located at the apical loop and flanking the stem, where they display a staggered symmetry (Fig. 4C). Importantly, the nucleotide sequences surrounding the 3′ cleavage site and the 3′ splice site are conserved in other vertebrates (Fig. 4B). Finally, fCLIP-seq experiments (Kim et al, 2017) showed that DROSHA specifically associates with spliced *SNHG25* transcript (Fig. 4B). Together, these analyses support our model by showing that the region predicted to fold in a stemloop is subject to endonucleolytic cleavage, and raise the possibility that the RNase III enzyme DROSHA is responsible for this processing step.

## Discussion

In this study, we have investigated how two co-hosted snoRNA can be matured from the same transcript precursor when one of them is intronic (snR107) (Leroy et al, 2025), and the other is not (snoU14). We demonstrated that the endoribonuclease Pac1 participates in the release of mature snoU14 from the precursor (Fig. 1). However, this activity is dispensable for snoU14 maturation (Fig. EV3), hinting at a secondary maturation pathway which likely involves another endoribonuclease. Similarly, in *S. cerevisiae* where snoU14 is released from a dicistronic precursor through cleavage by the RNase III ortholog Rnt1, partial functional redundancy between Rnt1 cleavage and another unknown factor has been reported (Chanfreau et al, 1998b). This raises the possibility that the homolog of this unknown factor also operates in *S. pombe* as part of a secondary snoU14 maturation pathway.

We identified that Pac1 cleaves the mamRNA-snoU14 precursor at a stemloop RNA structure encoded disjointly over its two exons (Fig. 2). The particular localization of this processing signal is remarkable, as it makes Pac1 activity specific towards the spliced version of the precursor transcript, delaying cleavage until splicing is complete. This unique mechanism explains the dramatic accumulation of mamRNA-snoU14 precursor previously reported

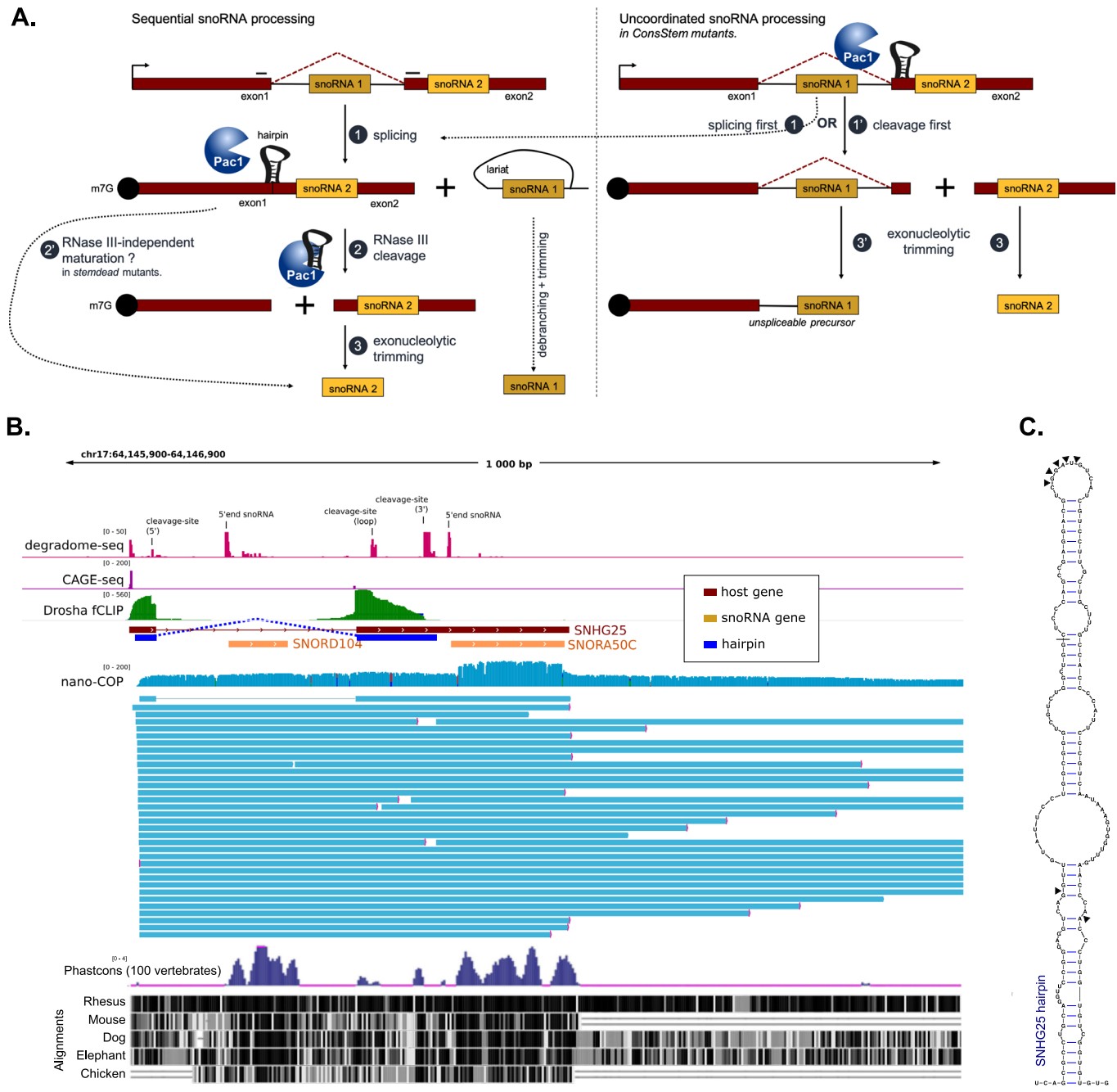

**Figure 4. Sequential snoRNA processing of mixed snoRNA clusters.**

(A) Model for sequential vs uncoordinated processing of co-hosted snoRNAs. In those models, m⁷G cap (black circles), polyA tails, and snoRNAs are barriers to exonucleolytic trimming. In sequential processing (left panel), splicing occurs first (1) allowing the release and maturation of the intronic snoRNA. By bringing together the two exons, splicing also allows for a secondary structure to fold across the exon–exon junction. This structure recruits the RNase III Pac1 in fission yeast (2), leading to the cleavage of the spliced precursor and subsequent maturation of the downstream snoRNA by exonucleolytic trimming (3). In the case of snoU14 in *S. pombe*, a secondary Pac1-independent maturation pathway exists (2'). In contrast, snoRNA processing would become uncoordinated (right panel) with cleavage potentially occurring before splicing (1') if the RNA secondary structure folded constitutively. This could lead to the exonucleolytic degradation of the 3' splice site, leaving the intronic snoRNA trapped in the precursor (3'). (B) Read density profiles and conservation score over the SNHG25 locus hosts to SNORD104 and SNORA50C snoRNAs. From top to bottom: degradome-seq and CAGE-seq (SRR1995152 and GSM7027460, respectively, where only the 5' end of reads were piled-up to improve resolution), DROSHA fCLIP-seq (GSE93651, where reads mapping to mature snoRNA, considered potential contaminant, were excluded for clarity), nano-COP data (long read sequencing for nascent RNA, GSE123191, for which representative individual reads are shown below the read density profile), and Phastcons conservation score (representing the probability of a nucleotide being part of a conserved element) calculated over 100 vertebrate species (Siepel et al, 2005) with pairwise multiZ alignments (Blanchette et al, 2004) between the SNHG25 gene and homologous loci in indicated species. (C) Predicted secondary structure from the sequences spanning the last exon–exon junction of SNHG25. Cleavage sites inferred by degradome-seq experiments are indicated by arrowheads.

**Table 1. Consequences of RNase III cleavage site localization relative to intron boundaries.**

| Localization | Consequences of cleavage | Shown in |
|---|---|---|
| Within intron | Quality control vs mis-spliced transcripts, regulation of targeted mRNA expression | *S. cerevisiae* (Rnt1) |
| Exon–intron junction | Competition with splicing, antagonizes splicing | Mammals (DROSHA) |
| Exon–exon junction | Splicing as a prerequisite to cleavage, delayed cleavage | *S. pombe* (Pac1) mammals? |

in splicing-deficient mutants (Leroy et al, 2025). We propose that requiring splicing as a prerequisite for Pac1 cleavage enables the precise control over the sequence of processing events. Indeed, we found that unconditional Pac1 cleavage of the mamRNA-snoU14 precursor leads to the accumulation of snR107-containing unspliced transcripts (Fig. 3), suggesting that premature cleavage occurring before splicing can antagonize the latter—possibly by exposing the 3′ splice site to degradation (Fig. 4A). It is likely that core snoRNP proteins bound to snR107 sequence (Leroy et al, 2025) can shield what remains of the precursor from further exonucleolytic trimming, similar to the lncRNA transcripts with snoRNA ends (sno-lncRNAs) described in mammals (Yin et al, 2012) and to hybrid messenger RNA–snoRNA (hmsnoRNA) described in *S. cerevisiae* (Liu et al, 2022). Therefore, we propose that enforcing delayed cleavage for snoU14 maturation ensures proper intronic snR107 maturation, which is critical to avoid the accumulation of a snR107-containing mamRNA isoform. Control over the expression of this isoform might be especially important, as both the mamRNA and snR107 contribute to the regulation of gametogenesis in fission yeast by interacting with the Mmi1 and Mei2 proteins (Leroy et al, 2025; Andric et al, 2021).

Previous studies have also described close mechanistic relationships between eukaryotic RNase III activity and intron splicing. In *S. cerevisiae*, Rnt1 cleavage sites are located within introns of seven pre-mRNAs coding for RNA-binding proteins, promoting the degradation of unspliced pre-mRNA and lariat introns (Gagnon et al, 2015; Danin-Kreiselman et al, 2003). In mammals, DROSHA cleavage sites span exon–intron junctions, antagonizing splicing either by direct cleavage or by sterically hindering splice-site recognition by the splicing machinery (Mattioli et al, 2013; Melamed et al, 2013; Havens et al, 2014; Lee et al, 2017). Our findings introduce a new, distinct configuration, where RNase III cleavage sites span exon–exon junctions and require prior splicing (Table 1). A similar case has been briefly reported among the non-canonical DROSHA substrates identified by genome-wide fCLIP-seq (formaldehyde cross-linking and immunoprecipitation) in human cell lines (Kim et al, 2017).

The enforced sequential snoRNA processing mechanism described here may have evolved to accommodate the unusual genomic organization of the two co-hosted snoRNAs from the mamRNA locus, where one snoRNA (snR107) depends on splicing for its biogenesis (Leroy et al, 2025), whereas the other (snoU14) is non-intronic and relies on endonucleolytic cleavage. Such an organization, which we classify as *polycistronic-mixed*, can be found in a wide range of eukaryotic species, though it is relatively uncommon compared to other types of snoRNA clusters (Fafard-Couture et al, 2024). However, its prevalence may be underestimated, as detecting polycistronic-mixed snoRNA clusters depends on high-quality snoRNA and host RNA annotation. Ongoing efforts to improve these annotations using machine learning and RNA sequencing optimized for snoRNA detection (Fafard-Couture et al, 2025) may enhance classification accuracy for snoRNA organization.

Nevertheless, the polycistronic-mixed snoRNA clusters identified in the current annotation were preferentially organized as a single non-intronic snoRNA located downstream of one or multiple intronic snoRNAs (Fig. EV4D). This consistent genomic organization, together with the presence of predictable stemloop structures folding across the last splice junction (Figs. 4C and EV4E), suggests that a cleavage signal spanning two exons may be a conserved mechanism to delay cleavage-dependent snoRNA maturation until after the last intronic snoRNA of the cluster has been spliced out. Since the mechanisms of 5′-end processing for non-intronic snoRNAs are unknown in higher eukaryotes, it remains to be determined which endonuclease(s) are involved (if any) and whether the predicted stemloops in the polycistronic-mixed clusters can indeed function as cleavage signals. Thus, the mechanisms underlying 5′-end processing of snoRNAs in eukaryotes remain a promising area for future investigation.

# Methods

**Reagents and tools table**

| Reagent/resource | Reference or source | Identifier or catalog number |
|---|---|---|
| **Experimental models** | | |
| *S. pombe* strains | This study | Table EV1 |
| **Recombinant DNA** | | |
| **Antibodies** | | |
| **Oligonucleotides and other sequence-based reagents** | | |
| PCR primer | This study | Table EV2 |
| Northern probes | This study | Table EV2 |
| **Chemicals, enzymes, and other reagents** | | |
| Rapamycin | Sigma | R8781 |
| Cordycepin | Sigma | C3394 |
| Acidic phenol-chloroform-isoamyl alcohol 125:24:1 | Sigma | P1944 |
| Chloroform-isoamyl alcohol 24:1 | Sigma | 25666 |
| DEPC | Sigma | D5758 |
| SYBR Green 2X Supermix | Bio-Rad | 172-5124 |
| Hybridization buffer | Sigma | H7033 |
| T4 polynucleotide kinase | Thermo Fisher Scientific | EK0031 |
| *E coli* polyA polymerase | NEB | M0276S |
| SuperScript III | Thermo Fisher Scientific | 18080093 |

| Reagent/resource | Reference or source | Identifier or catalog number |
|---|---|---|
| **Software** | | |
| Clustal Omega (EMBL-EBI Job Dispatcher) | Madeira et al, 2024 | |
| Alifold (web server version) | Hofacker, 2003 | |
| RNAfold (web server version) | Hofacker, 2003 | |
| vaRNA (v3.93) | Darty et al, 2009 | |
| Guppy (v6.5.7) | Oxford nanopore Technologies | |
| minimap2 | Li, 2018 | |
| IGV (v2.15.1) | Thorvaldsdottir et al, 2013 | |
| **Other** | | |
| High-Capacity cDNA Reverse Transcription Kit | Thermo Fisher Scientific | 4368813 |
| Amersham Hybond N+ membrane | Cytiva | RPN203B |

## Methods and protocols

### Yeast strains, drugs and media

Unless stated otherwise, yeast cells were grown at 32 °C in liquid minimal medium supplemented with adenine, uracil, histidine and leucine (EMM+AS) until an optical density of 0.4–0.6. For the anchor-away experiments, yeast cultures were treated for 2 h with 2.5 µg/ml rapamycin to allow for Pac1 nuclear exclusion, or with an equal volume of DMSO carrier. For the estimation of RNA decay rates, yeast were grown at 26 °C in EMM+AS over the full course of the experiment. Cordycepin (10 mM stock in $H_2O$, required heating at 55 °C to fully dissolve) was added to a final concentration of 0.6 mM at the appropriate time so that all sample from the same experiments could be pelleted together.

Gene disruptions and C-terminal tagging of proteins were performed by PCR-mediated gene targeting using the lithium acetate method. Cloning-free CRISPR/Cas9-mediated mutagenesis was used to generate the stemloop mutants (Zhang et al, 2018). Yeast strains used in this study are listed in Table EV1.

### About the choice of cordycepin as a transcriptional inhibitor

Chemical inhibition of transcription in yeast is most commonly performed by using a divalent cation chelator such as 1–10 phenanthroline or thiolutin (Qiu et al, 2024; Grigull et al, 2004). However, divalent cations are required for other processes beyond transcriptional elongation by the RNA polymerases; they are also important for the spliceosome to catalyze the second step of splicing (Shomron et al, 2002) as well as for Pac1 cleavage activity (Rotondo and Frendewey, 1996). Therefore, these transcriptional inhibitors interfere at different levels with the expression of the pre-mamRNA/snoU14, making them ill-suited to study its decay rate. As an alternative, we used the adenosine analog cordycepin. Compared to adenosine, cordycepin lacks an hydroxyl group required for chain elongation and therefore blocks transcription elongation when its triphosphorylated form is incorporated into nascent RNA (Horowitz et al, 1976).

### RNA extraction and analysis

Total RNA was extracted following the classical hot phenol method (Bähler and Wise, 2017) as previously described (Yague-Sanz et al, 2023). Briefly, the cells were first resuspended in 750 µl of TES solution (10 mM Tris-HCl [pH 7.5], 10mM EDTA [pH 8.0] and 0.5% SDS) and 750 of acidic phenol-chloroform-isoamyl alcohol 125:24:1 then incubated at 65 °C for 60 min with high-speed agitation for 10 seconds every 10 min. After centrifugation, the upper, aqueous, phase was transferred to a new tube containing equal volume of acidic phenol-chloroform-isoamyl alcohol 125:24:1. After thorough mixing and centrifugation, the upper phase was transferred to a new tube containing equal volume of chloroform-isoamyl alcohol 24:1. After mixing and centrifugation, 500 µL of the upper phase was precipitated with 1.5 mL 100% ethanol and 50 µL of 3M NaOAc pH5.2. The precipitated RNA pellet was washed twice with 70% ethanol, air-dried and resuspended in DEPC-treated water.

For RT-qPCR analysis, 0.5 µg of total RNA was retro-transcribed with the High-Capacity cDNA Reverse Transcription Kit following the manufacturer's instructions. Real-time PCR amplifications were performed with SYBR Green 2X Supermix in a Bio-Rad CFX96™ Real-Time machine. The PCR program was 3 min at 95 °C followed by 40 cycles of (15″ at 95 °C; 30″ at 60 °C) and a melt curve. The PCR reactions were performed in technical duplicates, and all experiments were performed at least in biological triplicates.

Relative RNA quantification relied on the $2^{-\Delta\Delta Cq}$ method using the actin gene as internal reference. For statistical analysis, the normalized relative expression levels were brought back in the Cq scale by log2 transformation and used in a one-sample Student's $t$ test. Primer sequences are listed in Table EV2.

For northern blot analysis, two types of denaturing gel were used: 1.5% agarose gel in MOPS buffer with 1.6M formaldehyde to separate (pre-)snoRNAs, as previously described (Andric et al, 2021), and 0.8% agarose gel in Tri/Tri buffer with 0.4M formaldehyde (Liu et al, 1999) to separate pre-rRNAs, as described (Duval et al, 2023). After migration, RNAs were transferred onto a positively charged nylon membrane (Amersham Hybond N+), crosslinked with UV in a Stratalinker or with heat (2h in the oven at 80 °C), and pre-hybridized in hybridization buffer at 42 °C. DNA probes were 5′ radiolabeled with [γ-32P]-ATP using T4 poly-nucleotide kinase (30 min at 37 °C) and were added to the membrane. After overnight incubation, membranes were washed once with 2 × SSC/0.1% SDS and twice with 0.1×SSC/0.1% SDS. A storage phosphor screen was exposed to the membrane for up to 24h and scanned in a Typhoon biomolecular imager (Amersham). When relevant, the samples were run in duplicates or triplicates on the same gel to allow the study of multiple probes without stripping. All northern blot experiments were performed at least in duplicates, with only one representative scan shown. Probe sequences are listed in Table EV2.

For 3′ RACE analysis, 10 µg of total RNA was polyadenylated using 5 U of *E coli* polyA polymerase during 20 min at 37 °C in the presence of RNase inhibitor. After phenol-chloroform purification, 1 µg of polyadenylated and non-polyadenylated total RNA were reversed-transcribed with SuperScript III reverse transcriptase from an oligo-dT primer according to the manufacturer's instructions. RNA was degraded with RNase H and the cDNA was used in PCR reactions with a gene-specific forward primer, a universal reverse

primer and TAQ polymerase (with 45 s of elongation and an annealing temperature of 61 °C). PCR products were ligated into pGEM-T easy vectors and electroporated into competent *E. coli* cells. Inserts were amplified by PCR and sequenced.

For long read sequencing direct RNA-seq polyA+ libraries (SQK-RNA002, Oxford Nanopore Technologies) were prepared according to the manufacturer's instructions starting from 500 ng of total RNA. The libraries were sequenced on a MinION apparatus during 48 h.

*Sequence analysis*

Clustal Omega accessed from the EMBL-EBI Job Dispatcher (Madeira et al, 2024) was used for multiple sequence alignment. RNA secondary structures were predicted using the Alifold and RNAfold2 webserver (Hofacker, 2003) and visualized in vaRNA (Darty et al, 2009). Short-read RNA-seq data and degradome-seq data were processed as previously described (Yague-Sanz et al, 2021; Zhang and Pelechano, 2021). For long read RNA-seq, the raw signal was basecalled with the Guppy algorithm with options --flowcell FLO-MIN106 --kit SQK-RNA002 --device auto --calib_-detect --reverse_sequence --u_substitution. The resulting reads were mapped onto *S. pombe* genome using minimap2 (Li, 2018), version 2.24, with options -Y -G 1000 -t 8 -R "@RG\tID:Sample\tSM:hs\tLB:ga\tPL:ONT" --MD -ax splice -uf -k14 --junc-bed and visualized in the IGV software (Thorvaldsdottir et al, 2013).

*Identification of mixed snoRNA clusters*

To identify mixed snoRNA clusters, we downloaded the latest genome annotation from Ensembl (Harrison et al, 2024) (version 110 released in July 2023) and selected snoRNAs co-hosted within the same precursor, with at least one being intronic and at least one being non-intronic. We filtered out ambiguous cases where snoRNAs overlapped exons defined as 'retained introns', reasoning that in these cases, the snoRNA is more likely to be intronic than not. Phastcons conservation score track (Siepel et al, 2005) and multiZ alignments (Blanchette et al, 2004) were downloaded from the UCSC genome browser.

# Data availability

The datasets produced or reanalyzed in this study are available in the following databases: Short-read RNA-seq data (*S. pombe*): GEO GSE167041; Degradome-seq data (*S. pombe*): ENA SRR12004691; Long read RNA-seq data (*S. pombe*): ENA PRJEB83877; Nano-cop data (*H. sapiens*): GEO GSE123191; CAGE-seq data (*H. sapiens*): GEO GSM7027460; degradome-seq data (*H. sapiens*): ENA SRR1995152; fCLIP-seq data (*H. sapiens*): GEO GSE93651.

The source data of this paper are collected in the following database record: biostudies:S-SCDT-10_1038-S44319-025-00553-y.

# Peer review information

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

## Acknowledgements

We thank Damien Hermand, Olivier De Backer, Marc Hennequart, Mathieu Rougemaille, Karine Choquet and François-Xavier Stubbe for helpful discussions and/or for proofreading the manuscript. We further thank Mathieu Rougemaille for generously sharing unpublished data regarding the existence of intronic snR107 in the mamRNA precursor. We also thank Damien Coupeau for his technical advice in setting up the 3' RACE experiments. Aligned nano-COP reads were kindly shared by Karine Choquet. CY-S is a scientific collaborator of the Fonds de la Recherche Scientifique – FNRS (grant number 40024367).

## Author contributions

**Valérie Migeot**: Investigation. **Yves Mary**: Investigation. **Etienne Fafard-Couture**: Data curation; Formal analysis; Investigation. **Pierre Lombard**: Investigation. **François Bachand**: Conceptualization; Writing—review and editing. **Michelle S Scott**: Conceptualization; Writing—review and editing. **Carlo Yague-Sanz**: Conceptualization; Data curation; Supervision; Investigation; Visualization; Methodology; Writing—original draft; Writing—review and editing.

Source data underlying figure panels in this paper may have individual authorship assigned. Where available, figure panel/source data authorship is listed in the following database record: biostudies:S-SCDT-10_1038-S44319-025-00553-y.

## Disclosure and competing interests statement

The authors declare no competing interests.

# Expanded View Figures

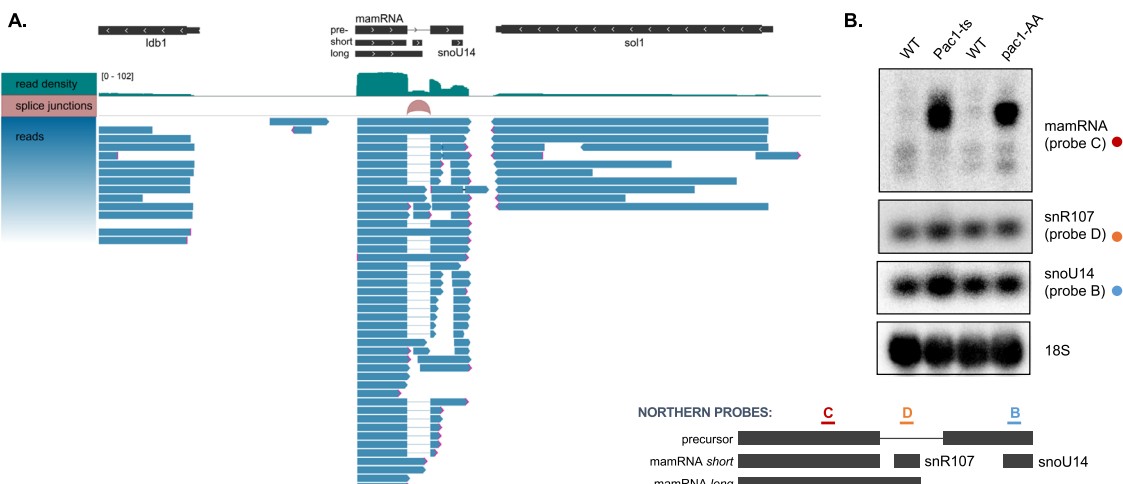

**Figure EV1.  A common precursor for mamRNA and snoU14 depends on Pac1 activity.**

(A) Read density, splice junctions and individual reads from a long read sequencing experiment (wild-type strain) over the mamRNA/snoU14 locus. (B) Representative northern blot analysis of transcripts originating from the mamRNA/snoU14 locus in conditions where the activity of Pac1 is compromised. Probes positions are indicated below (B).

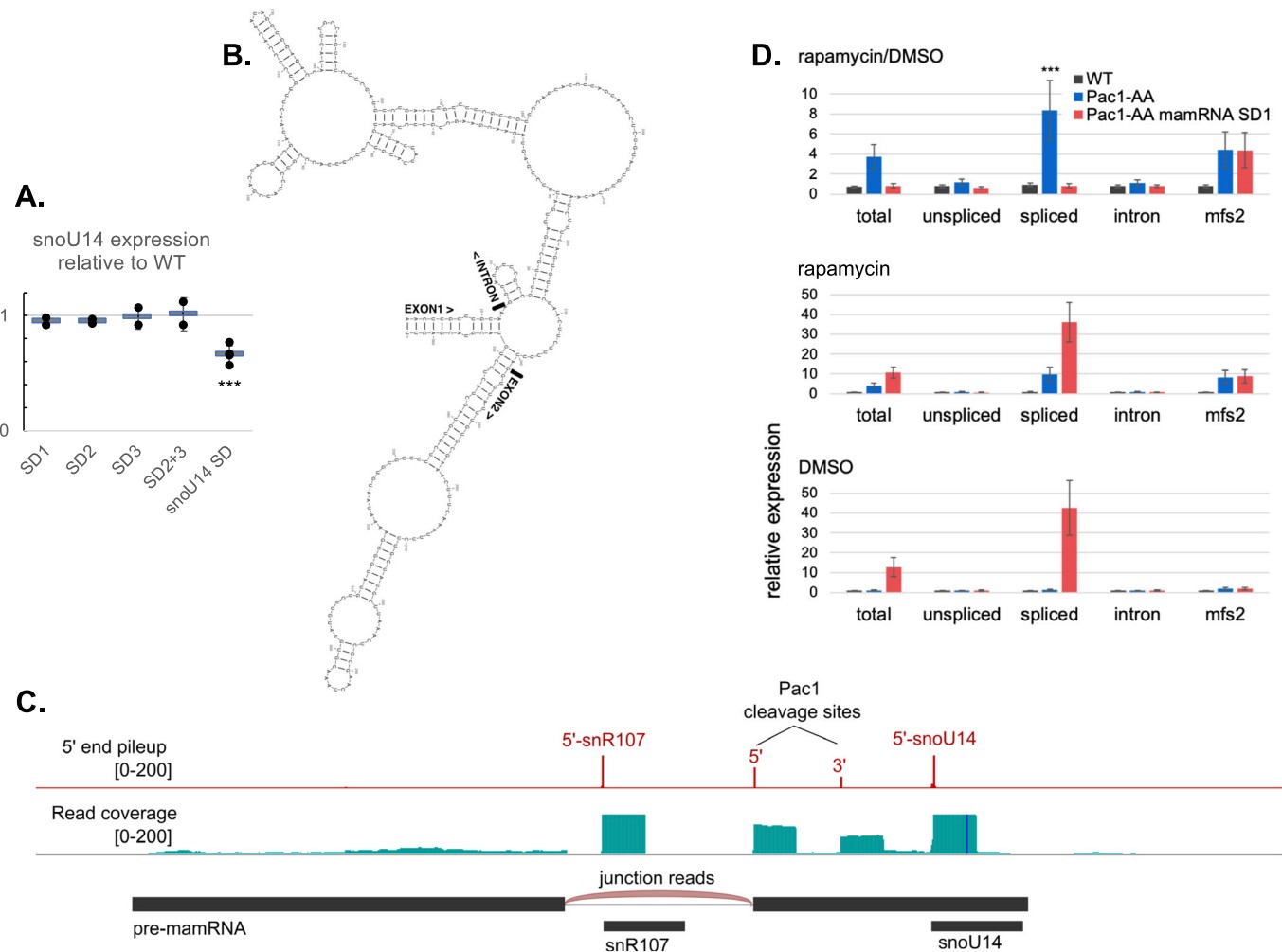

**Figure EV2. A stemloop spanning the exon–exon junction directs Pac1 cleavage at the mamRNA.**

(A) Semi-quantitative determination of mature snoU14 relative expression from pixel measurements of the northern blot experiments presented in Fig. 2B. Individual replicates (*n* = 4) are represented by filled circles. Means +/− SD are depicted by gray rectangle with error bars. Statistical significance of the differences in mature snoU14 level in the snoU14-SD mutant compared to the WT strains is indicated (***: Student's *t* test *P* value = 0.00046). (B) Predicted secondary structures of the unspliced pre-mamRNA/snoU14 transcript. (C) 5'-end pile-up of reads and overall coverage from a degradome-seq experiment in wild-type strain (SRR12004691 (Zhang and Pelechano, 2021)) over the mamRNA-snoU14 locus. The putative 5' and 3' Pac1 cleavage sites are indicated. (D) RT-qPCR analysis (*n* = 3) of transcript isoforms spanning the mamRNA locus in the indicated strains treated for 2 h with rapamycin (middle panel) or with its solvent (DMSO) as control (bottom panel). The qPCR amplicons are the same as in Fig. 1D with the addition of mfs2 as a positive control – mfs2 have been previously shown to be upregulated by conditional nuclear exclusion of Pac1 by rapamycin in the Pac1-AA strain (Yague-Sanz et al, 2021). The fold change and statistical significance of the rapamycin treatment effect compared to the DMSO control are indicated for the spliced pre-mamRNA isoform (top panel, ***: Student's *t* test *P* value = 0.00031). Error bars represent the standard deviation of the mean.

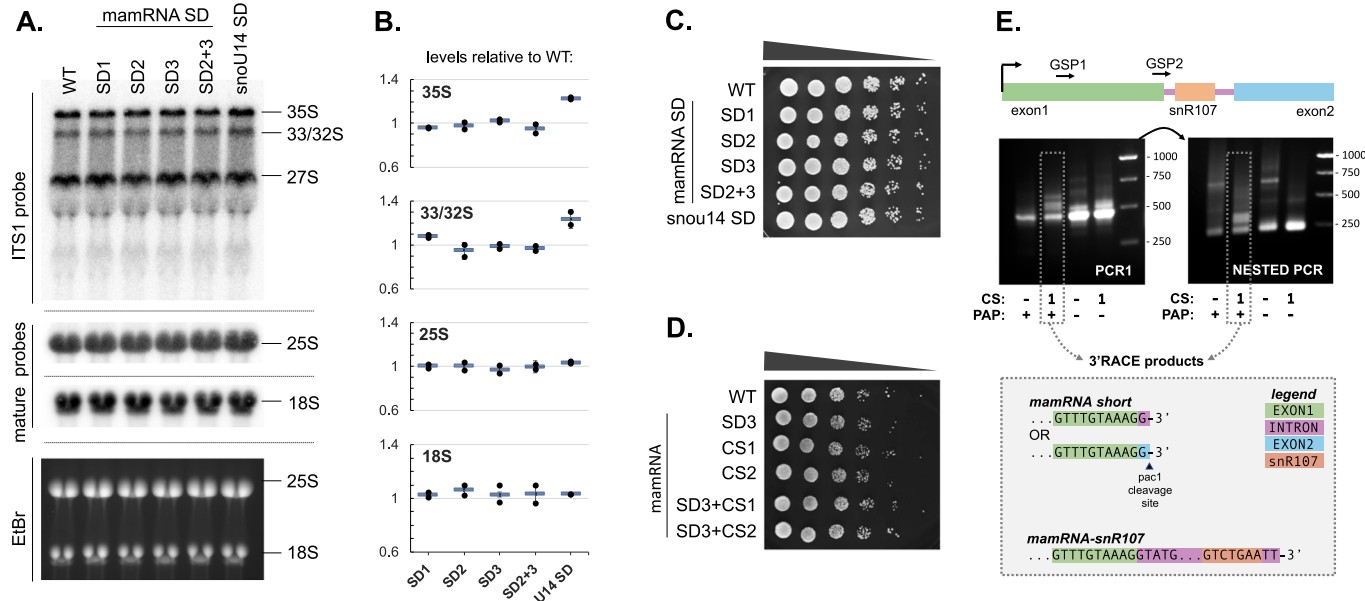

**Figure EV3.  snoU14 matured independently from Pac1 is functional.**

(A) Northern blot analysis of pre-rRNA transcripts in the indicated strains with internal transcribed spacer (ITS) probe. Mature rRNA forms (25S and 18S) are revealed by ethidium bromide (EthBr) staining. (B) Semi-quantitative determination of mature 35 and 33/32S rRNA precursor relative expression from pixel measurement of the northern blot experiments presented in (A). Individual replicates are represented by filled circles (n = 2). Their mean is depicted by a gray rectangle. (C) 5-fold dilutions of yeast cultures from the indicated strains spotted on YES-agar plates and incubated for 2 days at 32 °C. (D) 10-fold dilutions of yeast cultures from the indicated strains spotted on YES-agar plates and incubated for 3 days at 32 °C. (E) 3'-RACE analysis of mamRNA isoforms from total RNA extracted from CS1 mutant (CS) treated with poly(A) polymerase (PAP). Gene-specific primer (GSP) 1 and 2 were used in PCR1 and nested PCR, respectively. In this experiment, the mamRNA short isoforms ended one nucleotide shorter than currently annotated, with a G that could correspond either to the first nucleotide of the intron, or to the first nucleotide of exon2, in which case the 3'-end of the isoform matched the predicted 5' Pac1 cleavage site. Source data are available online for this figure.

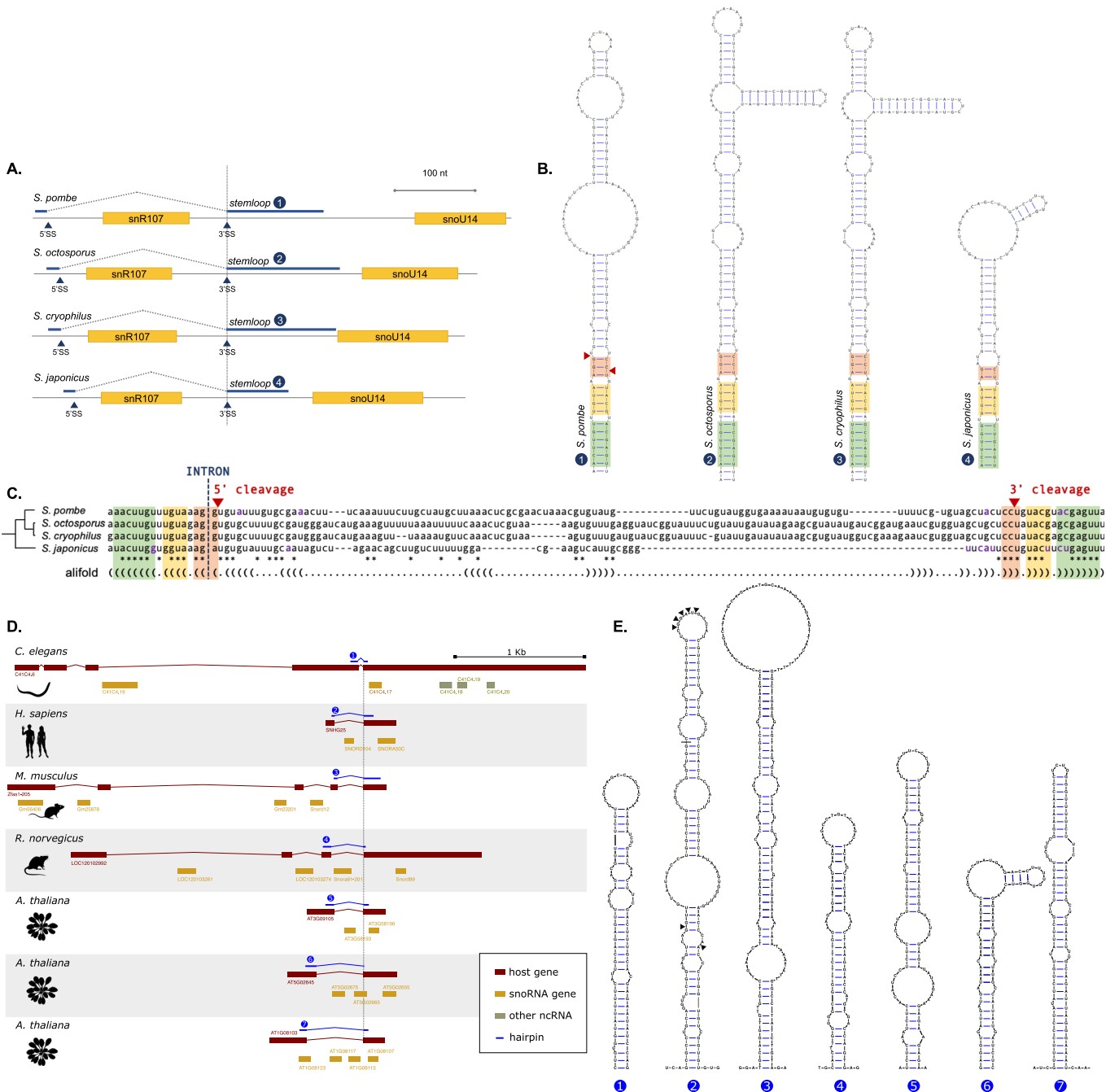

**Figure EV4. Conservation of secondary structures spanning splice junctions in mixed snoRNA clusters.**

(A) Genomic arrangements of snoU14 orthologs in *Schizosaccharomyces* species, aligned to their upstream 3′ splice site (3′SS). Unannotated snR107 orthologs were positioned in the upstream introns based on the conservation of C and D boxes, but with the exception of *Sp*snR107, their exact boundaries are uncertain. (B) Predicted secondary structure of the sequences spanning the exon–exon junction in (A). Arrowheads indicate the Pac1 cleavage site identified in *S. pombe*. Colored boxes indicate particularly conserved regions, as shown in (C). (C) Multiple sequence alignment of the predicted stem loops shown in (B). Conserved nucleotides are indicated by a star (*) and the consensus secondary structure of the alignment (alifold) is represented in dot-bracket notation. Red arrowheads indicate the Pac1 cleavage site identified in *S. pombe*. The exon–exon junction is indicated by a dotted blue line. Colored boxes indicate particularly conserved regions. Nucleotides that diverge from the consensus sequence while preserving the conserved structure are colored in purple. (D) Genomic arrangements of selected genes host to both intronic and non-intronic snoRNAs, aligned to the 5′ end of their last exon. (E) Predicted secondary structure from the sequences spanning the last exon–exon junction of host genes displayed in (B). Cleavage sites on SNHG25 inferred by degradome-seq experiments (stemloop #2) are indicated by arrowheads.

