## [Peer Review File · EMBO Reports]

RNase III cleavage sites spread across splice junctions enforce sequential snoRNA processing.

Valérie Migeot, Yves Mary, Étienne Fafard-Couture, Pierre Lombard, François Bachand, Michelle Scott, and Carlo Yague-Sanz

Corresponding author: Carlo Yague-Sanz (carlo.yaguesanz@unamur.be)

Review Timeline:

Transferred from Review Commons:	30th May 25
Editorial Decision:	7th Jul 25
Revision Received:	15th Jul 25
Accepted:	4th Aug 25

Editor: *Esther Schnapp*

Transaction Report:

Review
COMMONS

This manuscript was transferred to EMBO Reports following peer review at Review Commons.

Review #1**1. Evidence, reproducibility and clarity:****Evidence, reproducibility and clarity (Required)**

The authors describe a novel pattern of ncRNA processing by Pac1. Pac1 is a RNase III family member in *S. pombe* that has previously been shown to process pre-snoRNAs. Other RNase III family members, such as Rnt1 in *S. cerevisiae* and Dosha in human, have similar roles in cleaving precursors to ncRNAs (including miRNA, snRNA, snoRNA, rRNA). All RNase III family members share that they recognize and cleave dsRNA regions, but differ in their exact sequence and structure requirement. snoRNAs can be processed from their own precursor, a polycistronic pre-cursor, or the intron of a snoRNA host gene. After the intron is spliced out, the snoRNA host gene can either encode an protein or be a non-functional by product.

In the current manuscript the authors show that in *S. pombe* snoRNA snR107 and U14 are processed from a common precursor in a way that has not previously been described. snR107 is encoded within an intron and processed from the spliced out intron, similar to a typical intron-encoded snoRNA. What is different is that upon splicing, the host gene can adopt a new secondary structure that requires base-pairing between exon 1 and exon2, generating a Pac1 recognition site. This site is recognized, resulting in cleaving of the RNA and further processing of the 3' cleavage product into U14 snoRNA. In addition, the 5' cleavage product is processed into a ncRNA named mamRNA. The experiments describing this processing are thorough and convincing, and include RNAseq, degradome sequencing, northern blotting, qRT-PCR and the analysis of mutations that disrupt various secondary structures in figures 1, 2, and 3. The authors thereby describe a previously unknown gene design where both the exon and the intron are processed into a snoRNA. They conclude that making the formation of the Pac1 binding site dependent on previous splicing ensures that both snoRNAs are produced in the correct order and amount. Some of the authors findings are further confirmed by a different pre-print (reference 19), but the other reprint did not reveal the involvement of Pac1.

While the analysis on the mamRNA/snR107/U14 precursor is convincing, as a single example the impact of these findings is uncertain. In Figure 4 and supplemental table 1, the authors use bioinformatic searches and identify other candidate loci in plants and animals that may be processed similarly. Each of these loci encode a putative precursor that results in one snoRNA processed from an intron, a different snoRNA processed from an exon, and a double stranded structure that can only form after splicing. While is potentially

interesting, it is also the least developed and could be discussed and developed further as detailed below.

****Major comments:****

1. The proposal that plant and animal pre-snoRNA clusters are processed similarly is speculative. The authors provide no evidence that these precursors are processed by an RNase III enzyme cutting at the proposed splicing-dependent structure. This should not be expected for publication, but would greatly increase the interest.
2. The authors provide examples of similarly organized snoRNA clusters from human, mouse and rat, but the examples are not homologous to each other. Does this mean these snoRNA clusters are not conserved, even between mammals? Are the examples identified in *Arabidopsis* conserved in other plants? If there is no conservation, wouldn't that indicate that this snoRNA cluster organization offers no benefit?
3. Supplemental Figure 4 shows some evidence that the *S. pombe* gene organization is conserved within the *Schizosaccharomyces* genus, but could be enhanced further by showing what sequences/features are conserved. Presumably the U14 sequence is conserved, but snR107 is not indicated. Is it not conserved? Is the stem-loop more conserved than neighboring sequences? Are there any compensatory mutations that change the sequence but maintain the structure? Is there evidence for conservation outside the *Schizosaccharomyces* genus?
4. The authors suggest that snoRNAs can be processed from the exons of protein coding genes, but snoRNA processing would destroy the mRNA. Thus snoRNAs processing and mRNA function seem to be alternative outcomes that are mutually exclusive. Can the authors comment?

****Minor comments:****

5. The term "exonic snoRNA" is confusing. Isn't any snoRNA by definition an exon?
6. The methods section does not include how similar snoRNA clusters were identified in other species
7. In the discussion the authors argue that a previously published observation that *S. pombe* U14 does not complement a *S. cerevisiae* mutation can be explained because "was promoter elements... were simply not included in the transgene sequence". However, even if promoter elements were included, the dsRNA structure of *S. pombe* would not be cleaved by the *S. cerevisiae* RNase III. I doubt that missing promoter elements are the full explanation, and the authors provide insufficient data to support this conclusion.

****Referees cross-commenting****

I agree with the other 2 reviewers but think the thiouracil pulse labeling reviewer 2 suggests would take considerable work and if snoRNA processing is very fast might not be as conclusive as the reviewer suggests.

2. Significance:

Significance (Required)

In the current manuscript the authors show that in *S. pombe* snoRNA snR107 and U14 are processed from a common precursor in a way that has not previously been described. The experiments describing this processing are thorough and convincing, and include RNAseq, degradome sequencing, northern blotting, qRT-PCR and the analysis of mutations that disrupt various secondary structures in figures 1, 2, and 3. The authors thereby describe a previously unknown gene design where both the exon and the intron are processed into a snoRNA.

3. How much time do you estimate the authors will need to complete the suggested revisions:

Estimated time to Complete Revisions (Required)

(Decision Recommendation)

Less than 1 month

4. Review Commons values the work of reviewers and encourages them to get credit for their work. Select 'Yes' below to register your reviewing activity at Web of Science Reviewer Recognition Service (formerly Publons); note that the content of your review will not be visible on Web of Science.

No

Review #2

1. Evidence, reproducibility and clarity:

Evidence, reproducibility and clarity (Required)

The manuscript presents a novel mode of processing for polycistronic snoRNAs in the yeast *Saccharomyces pombe*. The authors demonstrate that the processing sequence of a

transcription unit containing U14, intronic snR107, and an overlapping non-coding mamRNA is determined by secondary structures recognized by RNase III (Pac1). Specifically, the formation of a stem structure over the mamRNA exon-exon junction facilitates the processing of terminal exonic-encoded U14. Consequently, U14 maturation occurs only after the mamRNA intron (containing snR107) is spliced out. This mechanism prevents the accumulation of unspliced, truncated mamRNA.

The first section describing the processing steps is challenging to follow due to the unusual organization of the locus and maturation pathway. If the manuscript is intended for a broad audience, I recommend simplifying this section and presenting it in a more accessible manner. A larger diagram illustrating the transcription unit and processing intermediates would be beneficial. Additionally, introducing snR107 earlier in the text would improve clarity.

Evaluation of some results is difficult due to the overexposure of Northern blot signals in Figures 1 and 2. The unspliced and spliced precursors appear as a single band, making it hard to distinguish processing intermediates. Would the authors consider presenting these results similarly to Figure 3, where bands are more clearly resolved? Or presenting both overexposed and underexposed blots?

Additionally, I noticed a discrepancy in U14 detection: Probe B gives a strong signal for U14 in Figure 3B, whereas in Figures 1 and 2, U14 appears as faint bands. Could the authors clarify this inconsistency? Furthermore, ethidium bromide (EtBr) staining of rRNA is used as a loading control, but overexposed signals from the gel may not accurately reflect RNA amounts on the membrane. This could affect the interpretation of mature RNA species' relative abundance.

To further support the sequential processing model, the authors could use pulse-labeling thioracil to test the accumulation of newly transcribed RNAs and accumulation of individual sopecies. Additionally, it could help determine whether U14 can be processed through alternative, less efficient pathways. Would the authors consider incorporating this approach?

In the final section, the authors propose that this processing mechanism is conserved across species, identifying 12 similar genetic loci in different organisms. This is very interesting finding. In my opinion, providing any experimental evidence would greatly strengthen this claim and the manuscript's significance. Even preliminary validation would add substantial value!

****Referees cross-commenting****

The other two reviewers' comments are justified.

2. Significance:

Significance (Required)

The authors describe an interesting novel mode of snoRNA processing from the host transcript. The results appear sound and intriguing, especially if the proposed mechanism can be confirmed across different organisms. Including such validation would significantly enhance the impact and make this work of broad audience interest.

My expertise: transcription, non-coding RNAs

3. How much time do you estimate the authors will need to complete the suggested revisions:

Estimated time to Complete Revisions (Required)

(Decision Recommendation)

Between 1 and 3 months

Yes

Review #3

1. Evidence, reproducibility and clarity:

Evidence, reproducibility and clarity (Required)

The manuscript by Migeot et al., focuses on a new Pac1-mediated snoRNA processing pathway for intron-encoded snoRNA pairs in yeast *Schizosaccharomyces pombe*.

The novelty of the findings described in MS is the report of an unusual and relatively rare

genomic organization and sequential processing of a few snoRNA genes in *S. pombe* and other eukaryotic organisms. It appears that in the case of snoRNA pairs, hosted in pre-mRNA in the intron and exon, respectively, the release of separate pre-snoRNAs from the host gene relies first on splicing to free the intron-encoded snoRNA, followed by endonucleolytic cleavage by RNase III (Pac1 in *S. pombe*) to produce snoRNA present in the mRNA exon. The sequential processing pathway, ensuring proper maturation of two snoRNAs, was demonstrated and argued in an elegant and clear way. The main message of the MS is straightforward, most experiments are properly conducted and specific conclusions based on the data are justified and valid. The text is clearly written and well-presented.

But there are some shortcomings. First of all, the title of the MS and general conclusions regarding the Pac1-mediated sequential release of snoRNA pairs hosted within the intron are definitely an overstatement. Especially the title suggests that this genomic organization and unusual processing mode of these snoRNAs is widespread. Later in the discussion the authors themselves admit that such mixed exonic-intronic snoRNAs are rare, although their presence may be underestimated due to annotation problems. It is likely that such snoRNA arrangement and processing is conserved, but the evidence is missing and only unique cases were identified based on bioinformatics mining and their processing has not been assayed. This makes the generalization impossible based on a single documented mamRNA/snoU14 example, no matter how carefully examined.

Another interesting observation is that, similarly to other intron-encoded snoRNA in other species, there is a redundant pathway to produce mature U14 in addition to Pac1-mediated cleavage. In the case of intronic snoRNAs in *S. cerevisiae*, their release could be performed either by splicing/debranching or Rnt1 cleavage, but there is also a third alternative option, that is processing following transcription termination downstream of the snoRNA gene, which at the same time interferes with the expression of the host gene. Is such a scenario possible as an alternative pathway for U14? Are there any putative, or even cryptic, terminators downstream of the U14 gene? The authors did not consider or attempt to inspect this possibility.

I also have some concerns or comments related to the presented research, which are not major, but are mainly related to data quantification, but have to be addressed.

In Pac1-ts and Pac1-AA strains the level of mature U14 seems upregulated compared to respective WT (Figure 1A). At the same time mature 25S and 18S rRNAs are less abundant. But there is no quantification and it is not mentioned in the text. What could be the reason for these effects?

Processing of the other snoRNA from the mamRNA/snoU14 precursor is largely overlooked

in the MS. It is commented on only in the context of mutants expressing constitutive mamRNA-CS constructs (Figure 3B). Its level was checked in Pac1-ts and Pac1-AA (Supplementary Figure 1), but the authors conclude that "its expression remained largely unaffected by Pac1 inactivation", which is clearly not true. Similarly to U14, also snR170 is increased in Pac1-ts and Pac1-AA strains, at least judged "by eye" because the loading control or quantification is not provided. This matter should be clarified.

Other minor comments:

****Minor points:****

1. Page 1, Abstract. The sentence "The hairpin recruits the RNase III Pac1 that cleaves and destabilizes the precursor transcript while participating in the maturation of the downstream exonic snoRNA, but only after splicing and release of the intronic snoRNA" is not entirely clear and should be simplified, maybe split into two sentences. This message is clear after reading the MS and learning the data, but not in the abstract.

2. Page 1, Introduction. I am not convinced by the need to use the term "exonic snoRNA" for all snoRNA that are not intronic, which is misleading, and is rather associated per se with snoRNA encoded in the mRNA exon. It has been used before in the review about snoRNAs by Michelle Scott published in RNA Biol (2024), but it does not justify its common use.

3. Supplementary Figure 3. It is difficult to assess whether the level of mature rRNAs is unchanged in the mutants based on EtBr staining and without calculations. Northern blotting should be performed and the levels properly calculated.

****Referees cross-commenting****

I also agree that 4sU labeling may require too much work with a questionable result.

2. Significance:

Significance (Required)

Strengths:

1. Novelty of the described genomic arrangement of snoRNA/ncRNA genes and their processing in a sequential and regulated manner.
2. Potential conservation of this pathways across eukaryotic organisms.
3. Well designed and performed experiments followed by proper conclusions.

Limitations:

1. Insufficient evidence to support generalization of the study results.
2. Moderate overall impact of the study

Advance:

This research can be placed within publications describing specific processing pathways for various non-coding RNAs, including for example unusual chimeric species such as sno-lncRNAs. In this context, the presented results do advance the knowledge in the field by providing mechanistic evidence for a tightly controlled and coordinated maturation of selected ncRNAs.

Audience:

Basic research and specialized. The interest in this research will rather be limited to a specific field.

3. How much time do you estimate the authors will need to complete the suggested revisions:

Estimated time to Complete Revisions (Required)

(Decision Recommendation)

Between 1 and 3 months

Yes

Full Revision

Manuscript number: RC-2025-02912

Corresponding author(s): Carlo, Yague-Sanz

1. General Statements [optional]

We thank the three reviewers for their time, effort, and constructive feedback, which have substantially contributed to improving the quality of our manuscript. We believe that we have thoroughly addressed all the points raised. In particular, we have expanded the section on the evolutionary conservation of the proposed mechanism, providing additional data and discussion that further support generalization our model.

To avoid confusion in the point-by-point response, we used a **black font** for the reviewer's comment and a **blue font** for our answer.

Reviewer #1 (Evidence, reproducibility and clarity (Required)):

The authors describe a novel pattern of ncRNA processing by Pac1. Pac1 is a RNase III family member in *S. pombe* that has previously been shown to process pre-snoRNAs. Other RNase III family members, such as Rnt1 in *S. cerevisiae* and Dosha in human, have similar roles in cleaving precursors to ncRNAs (including miRNA, snRNA, snoRNA, rRNA). All RNase III family members share that they recognize and cleave dsRNA regions, but differ in their exact sequence and structure requirement. snoRNAs can be processed from their own precursor, a polycistronic precursor, or the intron of a snoRNA host gene. After the intron is spliced out, the snoRNA host gene can either encode an protein or be a non-functional by product.

In the current manuscript the authors show that in *S. pombe* snoRNA snR107 and U14 are processed from a common precursor in a way that has not previously been described. snR107 is encoded within an intron and processed from the spliced out intron, similar to a typical intron-encoded snoRNA. What is different is that upon splicing, the host gene can adopt a new secondary structure that requires base-pairing between exon 1 and exon2, generating a Pac1 recognition site. This site is recognized, resulting in cleaving of the RNA and further processing of the 3' cleavage product into U14 snoRNA. In addition, the 5' cleavage product is processed into a ncRNA named mamRNA. The experiments describing this processing are thorough and convincing, and include RNAseq, degradome sequencing, northern blotting, qRT-PCR and the analysis of mutations that disrupt various secondary structures in figures 1, 2, and 3. The authors thereby describe a previously unknown gene design where both the exon and the intron are processed into a snoRNA. They conclude that making the formation of the Pac1 binding site

dependent on previous splicing ensures that both snoRNAs are produced in the correct order and amount. Some of the authors findings are further confirmed by a different pre-print (reference 19), but the other preprint did not reveal the involvement of Pac1.

While the analysis on the mamRNA/snR107/U14 precursor is convincing, as a single example the impact of these findings is uncertain. In Figure 4 and supplemental table 1, the authors use bioinformatic searches and identify other candidate loci in plants and animals that may be processed similarly. Each of these loci encode a putative precursor that results in one snoRNA processed from an intron, a different snoRNA processed from an exon, and a double stranded structure that can only form after splicing. While is potentially interesting, it is also the least developed and could be discussed and developed further as detailed below.

Major comments:

1. The proposal that plant and animal pre-snoRNA clusters are processed similarly is speculative. the authors provide no evidence that these precursors are processed by an RNase III enzyme cutting at the proposed splicing-dependent structure. This should not be expected for publication, but would greatly increase the interest.

All three reviewers expressed a similar concern, and we now provide additional evidence supporting the conservation of the proposed mechanism. Specifically, we focused on the *SNHG25* gene in *H. sapiens*, which hosts two snoRNAs—one intronic, as previously shown in Figure 4B, and one non-intronic. We substantiated our predictions through the re-analysis of multiple sequencing datasets in human cell lines, as outlined below:

I. Analysis of CAGE-seq and nano-COP datasets indicates a single major transcription initiation site at the *SNHG25* locus. Both the intronic and non-intronic snoRNAs are present within the same nascent precursor transcripts (Supplementary Figure 4D).

II. Degradome-seq experiments in human cell lines reveal that the predicted splicing-dependent stem-loop structure within the *SNHG25* gene is subject to endonucleolytic cleavage (Supplementary Figure 4D). The cleavage sites are located at the apical loop and flanking the stem, displaying a staggered symmetry characteristic of RNase III activity (Figure 4C). Importantly, the nucleotide sequence surrounding the 3' cleavage site and the 3' splice-site are conserved in other vertebrates (Supplementary Figure 4.D).

III. fCLIP experiments demonstrate that DROSHA associates with the spliced *SNHG25* transcript (Supplementary Figure 4D).

Together, these analyses support the generalizability of our model beyond fission yeast. They confirm the structure of the *SNHG25* gene as a single non-coding RNA precursor hosting two snoRNAs, one of which is intronic. Importantly, these findings show that the predicted stem-loop

structure contains conserved elements and is subject to endonucleolytic cleavage. Human DROSHA, an RNase III enzyme, could be responsible for this processing step.

2. The authors provide examples of similarly organized snoRNA clusters from human, mouse and rat, but the examples are not homologous to each other. Does this mean these snoRNA clusters are not conserved, even between mammals? Are the examples identified in Arabidopsis conserved in other plants? If there is no conservation, wouldn't that indicate that this snoRNA cluster organization offers no benefit?

We noticed during this revision that the human SNHG25 locus is actually very well conserved in mice at the GM36220 locus, where both snoRNAs (SNORD104 and SNORA50C/GM221711) are similarly arranged. Although the murine host gene, GM36220, also contains an intron in the UCSC annotation, it is intronless in the Ensembl annotation we used to screen for mixed snoRNA clusters, which explains why it was not part of our initial list of candidates (Supplementary Table 1). Importantly, sequence elements in SNHG25, close to the splice sites and cleavage sites in exon 2, are also well conserved in mice and other vertebrates (Supplementary Figure 4D). Therefore, it is reasonable to think that the mechanism described for SNHG25 in humans may also apply in mice and other vertebrates.

That being said, snoRNAs are highly mobile genetic elements. For example, it is well established that even between relatively closely related species (e.g., mouse and human), the positions of intronic snoRNAs within their host genes are not strictly conserved, even when both the snoRNAs and their host genes are. In the constrained drift model of snoRNA evolution (Hoepfner et al., BMC Evolutionary Biology, 2012; doi: 10.1186/1471-2148-12-183), it is proposed that snoRNAs are mobile and “*may occupy any genomic location from which expression satisfies phenotype.*”

Therefore, a low level of conservation in mixed snoRNA clusters is generally expected and does not necessarily imply that it offers no benefit. Despite the limited conservation of snoRNA identity across species, mixed snoRNA clusters consistently display two recurring features: (1) non-intronic snoRNAs often follow intronic snoRNAs, and (2) the predicted secondary structure tends to span the last exon–exon junction. These enriched features support the idea that enforcing sequential processing of mixed snoRNA clusters may confer a selective advantage. We now explicitly discuss these points in the revised manuscript.

3. Supplemental Figure 4 shows some evidence that the *S. pombe* gene organization is conserved within the Schizosaccharomyces genus, but could be enhanced further by showing what sequences/features are conserved. Presumably the U14 sequence is conserved, but snR107 is not indicated. Is it not conserved? Is the stem-loop more conserved than neighboring sequences? Are there any compensatory mutations that change the sequence but maintain the structure? Is there evidence for conservation outside the Schizosaccharomyces genus?

We thank the reviewer for these excellent suggestions, which helped us significantly improve Supplementary Figure 4. In the revised version, we now include an additional species—*S. japonicus*, which is more evolutionarily distant—and show that the intronic *snR107* is conserved across the *Schizosaccharomyces* genus (Supplementary Figure 4A). The distance between conserved elements (splice sites, snoRNAs, and RNA structures) varies, indicating that surrounding sequences are less conserved compared to these functionally constrained features

We also performed a detailed alignment of the sequences corresponding to the predicted RNA secondary structures. This revealed that the apical regions are less conserved than the base, particularly near the splice and cleavage sites. In these regions, we observe compensatory or base-pair-neutral mutations (e.g., U-to-C or C-to-U, which both pair with G), suggesting structural conservation through evolutionary constraint (Supplementary Figures 4B–C). These observations are now described in greater detail in the revised manuscript, along with a discussion of the specific features likely to be under selective pressure at this locus.

Conservation outside the *Schizosaccharomyces* genus is less clear. As already noted in the manuscript, the *S. cerevisiae* locus retains synteny between *snR107* and *snoU14*, but the polycistronic precursor encompassing both is intronless and processed by RNase III (Rnt1) between the cistrons. Similarly, in *Ashbya gossypii* and a few other fungal species, synteny is preserved, but no intron appears to be present in the presumed common precursor. Notably, secondary structure predictions for the *A. gossypii* locus (not shown) suggest the formation of a stable stem-loop encompassing the first snoRNA in a large apical loop. This could reflect a distinct mode of snoRNA maturation, possibly analogous to pri-miRNA processing, where cleavage by an RNase III enzyme contributes to both 5' and 3' end formation. In *Candida albicans*, *snoU14* is annotated within an intron of a host gene, but no homolog of *snR107* is annotated. Other cases either resemble one of the above scenarios or are inconclusive due to the lack of a clearly conserved snoRNA (or possibly due to incomplete annotation). Although these examples are potentially interesting, we have chosen not to elaborate on them in the manuscript in order to maintain focus and avoid speculative interpretation in the absence of stronger evidence.

4. The authors suggest that snoRNAs can be processed from the exons of protein coding genes, but snoRNA processing would destroy the mRNA. Thus snoRNAs processing and mRNA function seem to be alternative outcomes that are mutually exclusive. Can the authors comment?

In theory, we agree with reviewer on the mutually exclusive nature of mRNA and snoRNA expression for putative snoRNA hosted in the exon of protein coding genes. However, we want to clarify that the specific examples of snoRNA precursor (or host) developed in the manuscript (mamRNA-snoU14 in *S. pombe* and, in this resubmission, SNHG25 in *H. sapiens*) are non-coding. So although we do not exclude that our model of sequential processing through splicing and endonucleolytic cleavage could apply to coding snoRNA precursors, it is not something we want to insist on, especially given the lack of experimental evidence for these cases.

It is possible that the use of the term "*exonic snoRNA*" in the first version of the manuscript led to the reviewer's impression that we explicitly meant that snoRNA processing can be processed from the exon of protein coding genes, which was not what we meant (although we do not exclude it). If that was the case, we apologize for the confusion. We have now clarified the issue (see next point).

Minor comments:

5. The term "exonic snoRNA" is confusing. Isn't any snoRNA by definition an exon?

We agree that this term can be confusing, a sentiment that was also shared by reviewer 3. We replaced the problematic term by either "non-intronic snoRNA", "snoRNA" or "snoRNA gene located in exon" depending on the context, which are more unambiguous in conveying our intended meaning.

6. The methods section does not include how similar snoRNA clusters were identified in other species

We have now corrected this omission in the method section ('Identification of mixed snoRNA clusters' subsection): "To identify mixed snoRNA clusters, we downloaded the latest genome annotation from Ensembl and selected snoRNAs co-hosted within the same precursor, with at least one being intronic and at least one being non-intronic. We filtered out ambiguous cases where snoRNAs overlapped exons defined as 'retained introns', reasoning that in these cases the snoRNA is more likely to be intronic than not."

7. In the discussion the authors argue that a previously published observation that *S. pombe* U14 does not complement a *S. cerevisiae* mutation can be explained because "was promoter elements... were simply not included in the transgene sequence". However, even if promoter elements were included, the dsRNA structure of *S. pombe* would not be cleaved by the *S. cerevisiae* RNase III. I doubt that missing promoter elements are the full explanation, and the authors provide insufficient data to support this conclusion.

We agree with the reviewer that, given the substantial divergence in substrate specificity between Pac1 and Rnt1, it is unlikely that *S. pombe* snoU14 would be efficiently processed from its precursor in *S. cerevisiae*. We did not intend to suggest otherwise, and we have now removed this part of the discussion. As the experiment reported by Samarsky et al. did not detect expression of the *S. pombe* snoU14 precursor (even its unprocessed form), it remains inconclusive with respect to the conservation (or lack thereof) of snoU14 processing mechanisms.

Full Revision

For the record, we had originally included this discussion to point out that the lack of cryptic promoter activity (or at least none that *S. cerevisiae* can use) within the *S. pombe* snoU14 precursor supports the idea that transcription initiates solely upstream of the mamRNA precursor. However, we recognize that this argument is speculative and potentially confusing. We have therefore removed it from the revised manuscript to maintain clarity and focus.

****Referees cross-commenting****

I agree with the other 2 reviewers but think the thiouracil pulse labeling reviewer 2 suggests would take considerable work and if snoRNA processing is very fast might not be as conclusive as the reviewer suggests.

We are grateful to the reviewer for this comment, which helped us perform this reviewing in a timely manner.

Reviewer #1 (Significance (Required)):

In the current manuscript the authors show that in *S. pombe* snoRNA snR107 and U14 are processed from a common precursor in a way that has not previously been described. The experiments describing this processing are thorough and convincing, and include RNAseq, degradome sequencing, northern blotting, qRT-PCR and the analysis of mutations that disrupt various secondary structures in figures 1, 2, and 3. The authors thereby describe a previously unknown gene design where both the exon and the intron are processed into a snoRNA.

Reviewer #2 (Evidence, reproducibility and clarity (Required)):

The manuscript presents a novel mode of processing for polycistronic snoRNAs in the yeast *Saccharomyces pombe*. The authors demonstrate that the processing sequence of a transcription unit containing U14, intronic snR107, and an overlapping non-coding mamRNA is determined by secondary structures recognized by RNase III (Pac1). Specifically, the formation of a stem structure over the mamRNA exon-exon junction facilitates the processing of terminal exonic-encoded U14. Consequently, U14 maturation occurs only after the mamRNA intron (containing snR107) is spliced out. This mechanism prevents the accumulation of unspliced, truncated mamRNA.

1. The first section describing the processing steps is challenging to follow due to the unusual organization of the locus and maturation pathway. If the manuscript is intended for a broad audience, I recommend simplifying this section and presenting it in a more accessible manner. A larger diagram illustrating the transcription unit and processing intermediates would be beneficial. Additionally, introducing snR107 earlier in the text would improve clarity.

We thank the reviewer for these excellent suggestions. In the previous version of the manuscript, we were cautious in how we introduced the locus, as *snR107* and the associated intron had not yet been published. This is no longer the case, as the locus is now described in Leroy et al. (2025). Accordingly, we now introduce the complete locus at the beginning of the manuscript and have improved the corresponding diagram (new Figure 1A). We believe these changes enhance clarity and make the section more accessible to a broader audience.

3. Evaluation of some results is difficult due to the overexposure of Northern blot signals in Figures 1 and 2. The unspliced and spliced precursors appear as a single band, making it hard to distinguish processing intermediates. Would the authors consider presenting these results similarly to Figure 3, where bands are more clearly resolved? Or presenting both overexposed and underexposed blots?

For all blots (probes A, B, and C), we selected an exposure level that allows detection of precursor forms under wild-type (WT) conditions. This necessarily results in some overexposure of the accumulating precursors in mutant conditions, due to their broad dynamic range of accumulation. To address this, we now provide an additional supplementary "source data" file containing all uncropped blots with both low and high exposures.

For example, a lower exposure version of the blot in new Figure 1.B (included below) confirms the consistent accumulation of the spliced precursor when Pac1 activity is compromised. The unspliced precursor also shows slight accumulation in the *Pac1-ts* mutant, although to a much lesser extent than the spliced precursor. This observation is consistent with our qPCR results (new Figure 1.C).

Importantly, because this effect is not observed in neither the *Pac1-AA* or the steam-dead (SD) mutants, we interpret it as an indirect effect—possibly reflecting a mild growth defect in the *Pac1-ts* strain, even under growth-permissive conditions. We now explicitly address this point in the revised manuscript.

Lower exposure for Northern blot in figure 1.B. RNA from WT, pac1-ts, CTL, Pac1-AA loaded 3 times in this order, probed with probe A (left), B (middle) and C (right).

4. Additionally, I noticed a discrepancy in U14 detection: Probe B gives a strong signal for U14 in Figure 3B, whereas in Figures 1 and 2, U14 appears as faint bands. Could the authors clarify this inconsistency?

We thank the reviewer for pointing out this discrepancy. The variation in U14 signal intensity is most likely due to technical differences in UV crosslinking efficiency during the Northern blot procedure. This step can differentially affect the membrane retention of RNA species depending on their length, as previously reported (PMID: 17405769). Because U14 is a relatively abundant snoRNA, the fainter signal observed in Figure 1 (relative to the accumulating precursor) likely reflects suboptimal crosslinking of shorter RNAs in that particular blot.

Importantly, this technical variability does not impact the conclusions of our study, as we do not compare RNA species of different lengths directly. To increase transparency, we now provide a supplementary "source data" file that includes all uncropped blots from our Northern blot experiments. These include examples—such as the blot shown below under the same conditions as Figure 1B—where U14 retention is more consistent.

Uncropped blot for Northern blot in S1 (same conditions than in figure 1.B). RNA from WT, *pac1-ts*, CTL, *Pac1-AA*. Probe B.

5. Furthermore, ethidium bromide (EtBr) staining of rRNA is used as a loading control, but overexposed signals from the gel may not accurately reflect RNA amounts on the membrane. This could affect the interpretation of mature RNA species' relative abundance.

We thank the reviewer for pointing this out and have now measured rRNAs loading on the same northern blot membrane from probes complementary to mature rRNA. We updated new Figures 1B, 2B, 3B, S1B, and S3A accordingly.

6. To further support the sequential processing model, the authors could use pulse-labeling thioracil to test the accumulation of newly transcribed RNAs and accumulation of individual species. Additionally, it could help determine whether U14 can be processed through alternative, less efficient pathways. Would the authors consider incorporating this approach?

We thank the reviewer for this pertinent suggestion. We actually plan to investigate the putative alternative U14 maturation pathway in future work, and the suggested approach will definitely be instrumental for that. However, to keep the present manuscript focused, and also to keep the review timely (successful pulse-chase experiments are likely to take time to optimize – as also suggested by the other reviewers in their cross-commenting section), we prefer not to perform this experiment for this reviewing.

7. In the final section, the authors propose that this processing mechanism is conserved across species, identifying 12 similar genetic loci in different organisms. This is very interesting finding. In my opinion, providing any experimental evidence would greatly strengthen this claim and the manuscript's significance. Even preliminary validation would add substantial value!

We thank the reviewer for his/her enthusiasm and are glad to provide some preliminary validation to the final section of our manuscript. Specifically, we focused on the *SNHG25* gene in *H. sapiens*,

which hosts two snoRNAs—one intronic, as previously shown in Figure 4B, and one non-intronic. We substantiated our predictions through the re-analysis of multiple sequencing datasets in human cell lines, as outlined below:

I. Analysis of CAGE-seq and nano-COP datasets indicates a single major transcription initiation site at the *SNHG25* locus. Both the intronic and non-intronic snoRNAs are present within the same nascent precursor transcripts (Supplementary Figure 4D).

II. Degradome-seq experiments in human cell lines reveal that the predicted splicing-dependent stem-loop structure within the *SNHG25* gene is subject to endonucleolytic cleavage (Supplementary Figure 4D). The cleavage sites are located at the apical loop and flanking the stem, displaying a staggered symmetry characteristic of RNase III activity (Figure 4C). Importantly, the nucleotide sequence surrounding the 3' cleavage site and the 3' splice-site are conserved in other vertebrates (Supplementary Figure 4.D).

III. fCLIP experiments demonstrate that DROSHA associates with the spliced *SNHG25* transcript (Supplementary Figure 4D).

Together, these analyses support the generalizability of our model beyond fission yeast. They confirm the structure of the *SNHG25* gene as a single non-coding RNA precursor hosting two snoRNAs, one of which is intronic. Importantly, these findings unambiguously show that the predicted stem-loop structure is subject to endonucleolytic cleavage, and they are consistent with DROSHA, an RNase III enzyme, being responsible for this processing step.

****Referees cross-commenting****

The other two reviewers' comments are justified.

Reviewer #2 (Significance (Required)):

The authors describe an interesting novel mode of snoRNA processing from the host transcript. The results appear sound and intriguing, especially if the proposed mechanism can be confirmed across different organisms. Including such validation would significantly enhance the impact and make this work of broad audience interest.

My expertise: transcription, non-coding RNAs

Reviewer #3 (Evidence, reproducibility and clarity (Required)):

The manuscript by Migeot et al., focuses on a new Pac1-mediated snoRNA processing pathway for intron-encoded snoRNA pairs in yeast *Schizosaccharomyces pombe*. The novelty of the findings described in MS is the report of an unusual and relatively rare genomic organization and sequential processing of a few snoRNA genes in *S. pombe* and other eukaryotic organisms. It appears that in the case of snoRNA pairs, hosted in pre-mRNA in the intron and exon, respectively, the release of separate pre-snoRNAs from the host gene relies first on splicing to free the intron-encoded snoRNA, followed by endonucleolytic cleavage by RNase III (Pac1 in *S. pombe*) to produce snoRNA present in the mRNA exon. The sequential processing pathway, ensuring proper maturation of two snoRNAs, was demonstrated and argued in an elegant and clear way. The main message of the MS is straightforward, most experiments are properly conducted and specific conclusions based on the data are justified and valid. The text is clearly written and well-presented.

But there are some shortcomings.

1. First of all, the title of the MS and general conclusions regarding the Pac1-mediated sequential release of snoRNA pairs hosted within the intron are definitely an overstatement. Especially the title suggests that this genomic organization and unusual processing mode of these snoRNAs is widespread. Later in the discussion the authors themselves admit that such mixed exonic-intronic snoRNAs are rare, although their presence may be underestimated due to annotation problems. It is likely that such snoRNA arrangement and processing is conserved, but the evidence is missing and only unique cases were identified based on bioinformatics mining and their processing has not been assayed. This makes the generalization impossible based on a single documented mamRNA/snoU14 example, no matter how carefully examined.

We thank the reviewer for clearly articulating this concern. In response, we now provide additional evidence supporting conservation of the proposed mechanism in other species:

- Conservation within the *Schizosaccharomyces* genus (Figures S4A–C) has been further analyzed, as suggested by Reviewer 1. This expanded analysis highlights conserved features—such as splice sites and cleavage sites within the predicted stem-loop structure—indicating that these elements are under selective constraint.

- Conservation in mammals is now supported by experimental data, as detailed in our responses to point #7 of Reviewer 2 and major comment #1 of Reviewer 1. Specifically, we show that for the *SNHG25* gene in *H. sapiens* (Figure S4D):

- (1) nascent transcription give rise to a single non-coding RNA precursor that hosts two snoRNAs, one of which is intronic;
- (2) the predicted stem-loop structure contains conserved elements and is subject to endonucleolytic cleavage;
- (3) the RNase III enzyme DRISHA associates with the spliced *SNHG25* precursor.

Together, these analyses strengthen the evidence for the evolutionary conservation of the mechanism and support the general conclusions and title of the manuscript.

2. Another interesting observation is that, similarly to other intron-encoded snoRNA in other species, there is a redundant pathway to produce mature U14 in addition to Pac1-mediated cleavage. In the case of intronic snoRNAs in *S. cerevisiae*, their release could be performed either by splicing/debranching or Rnt1 cleavage, but there is also a third alternative option, that is processing following transcription termination downstream of the snoRNA gene, which at the same time interferes with the expression of the host gene. Is such a scenario possible as an alternative pathway for U14? Are there any putative, or even cryptic, terminators downstream of the U14 gene? The authors did not consider or attempt to inspect this possibility.

We thank the reviewer for this interesting and thoughtful comment. First, we would like to clarify that *snoU14* is not intron-encoded; rather, it is located on the exon downstream of the intron-encoded *snR107*.

Regarding the possibility of transcription termination-based processing: downstream of *snoU14*, we identified a non-consensus polyadenylation signal (AUUAAA) preceded by a U-rich tract, followed by three consensus polyadenylation signals (AAUAAA) within a 500-nt window. These elements likely contribute to robust and redundant transcription termination at this highly expressed locus. However, since all these sites are located downstream of *snoU14*, they do not provide an alternative 5'-end processing mechanism for this snoRNA –they reflect normal termination.

If we correctly understood the reviewer's suggestion (apologies if not), they may have been referring to the possibility of a cryptic or alternative polyadenylation site **between** *snR107* and *snoU14* instead. If cleavage were to occur in this inter-snoRNA region while transcription continued past *snoU14*, it could, in principle, allow for alternative processing of *snoU14*. We have indeed considered this scenario. However, we currently do not find strong support for it: there are no identifiable polyadenylation signals motifs between the two snoRNAs, aside from a weakly conserved and questionable AAUAAU hexamer that does not appear to be used as polyA site at least in WT conditions (DOI: 10.4161/rna.25758). Given the lack of evidence, we chose not to explore this hypothesis further in the present manuscript, though it remains an interesting possibility for future investigation.

3. I also have some concerns or comments related to the presented research, which are no major, but are mainly related to data quantification, but have to be addressed.

- In Pac1-ts and Pac1-AA strains the level of mature U14 seems upregulated compared to respective WT (Figure 1A). At the same time mature 25S and 18S rRNAs are less abundant. But there is no quantification and it is not mentioned in the text. What could be the reason for these effects?

We thank the reviewer for this observation. As reviewer 2 also noted, ethidium bromide staining of mature rRNAs is not a reliable quantitative loading control. In response to this concern, we have now reprobbed all northern blots with radiolabeled rRNA probes. These provide a more accurate and consistent loading for our blots (new Figures 1B, 2B, 3B, S1B, S3A).

Using these improved loading controls, it is evident that *snoU14*, *snR107*, and the unspliced precursor are all slightly upregulated in the Pac1-ts strain, although to a much lesser extent than the spliced precursor, which accumulates dramatically. We do not observe this effect in either the Pac1-AA or stem-dead (SD) mutants. We therefore interpret the modest upregulation as an indirect effect, possibly linked to the physiological state of the Pac1-ts mutant, which exhibits slower growth even at growth-permissive temperatures. We now explicitly discuss this in the revised manuscript.

Regarding the suggestion to include quantification of the northern blot signal: we opted not to include this in the figures for the following reasons. First, the accumulation of the spliced precursor—the central focus of our analysis—is large and highly reproducible across all replicates and conditions. Second, northern blot quantification by pixel intensity remains semi-quantitative, particularly for comparisons across RNAs of highly different abundance. Finally, we support our conclusions with additional quantitative data from RT-qPCR and RNA-seq, which provide more robust measures of RNA accumulation.

- Processing of the other snoRNA from the *mamRNA*/*snoU14* precursor is largely overlooked in the MS. It is commented on only in the context of mutants expressing constitutive *mamRNA*-CS constructs (Figure 3B). Its level was checked in Pac1-ts and Pac1-AA (Supplementary Figure 1), but the authors conclude that "its expression remained largely unaffected by Pac1 inactivation", which is clearly not true. Similarly to U14, also *snR170* is increased in Pac1-ts and Pac1-AA strains, at least judged "by eye" because the loading control or quantification is not provided. This matter should be clarified.

We thank the reviewer for pointing this out. We have now included appropriate loading controls for Supplementary Figure 1 to clarify the interpretation. As discussed in our response to the previous comment, we observe a general upregulation of the *mamRNA* locus in the Pac1-ts strain, which likely contributes to the increased levels of both *snR107* and *snoU14*. However, because this upregulation is not observed in the Pac1-AA or stem-dead (SD) mutants, we interpret it as an indirect effect, possibly related to the altered physiological state of the Pac1-ts strain (*e.g.*, slightly

reduced growth rate even at the permissive temperature). This interpretation has now been clearly explained in the revised manuscript.

We also identified and corrected a labeling error in the previous version of Supplementary Figure 1, where the Pac1-ts and Pac1-AA strains were inadvertently swapped. We sincerely apologize for the confusion this may have caused and have now ensured that all figure panels are correctly labeled and consistent with the text.

Other minor comments:

Minor points:

1. Page 1, Abstract. The sentence "The hairpin recruits the RNase III Pac1 that cleaves and destabilizes the precursor transcript while participating in the maturation of the downstream exonic snoRNA, but only after splicing and release of the intronic snoRNA" is not entirely clear and should be simplified, maybe split into two sentences. This message is clear after reading the MS and learning the data, but not in the abstract.

We thank the reviewer for pointing this out and have now clarified the abstract following the suggestion to split and simplify the problematic sentence : "*... the sequence surrounding an exon-exon junction within their precursor transcript folds into a hairpin after splicing of the intron. This hairpin recruits the RNase III ortholog Pac1, which participates in the maturation of the downstream snoRNA by cleaving the precursor.*"

2. Page 1, Introduction. I am not convinced by the need to use the term "exonic snoRNA" for all snoRNA that are not intronic, which is misleading, and is rather associated per se with snoRNA encoded in the mRNA exon. It has been used before in the review about snoRNAs by Michelle Scott published in RNA Biol (2024), but it does not justify its common use.

We thank the reviewer for raising this important point. We agree that the term "exonic snoRNA" can be misleading, as it was previously used to specifically refer to snoRNAs embedded within exons of mRNA transcripts—an rare and potentially artifactual scenario, as very cautiously discussed by Michelle Scott and colleagues in their review published in RNA Biol (2024).

In the previous version of our manuscript, we actually used "exonic snoRNA" in a broader sense to denote any snoRNA not encoded within an intron, primarily for convenience in contrasting the processing of *intronic* snR107 with that of *non-intronic/exonic* snoU14. However, we recognize that this usage is non-standard and risks confusion due to the ambiguity surrounding the term's definition in the literature.

Full Revision

In light of this, and in agreement with reviewer 1 who raised a similar concern, we have revised the manuscript to remove the term “exonic snoRNA” entirely. Depending on the context, we now refer more precisely to “non-intronic snoRNA,” “snoRNA gene located in exon,” or simply “snoRNA.”

3. Supplementary Figure 3. It is difficult to assess whether the level of mature rRNAs is unchanged in the mutants based on EtBr staining and without calculations. Northern blotting should be performed and the levels properly calculated.

As suggested, we performed northern blotting on mature 18S and 25S, quantified the signal and observed no significant differences (new Supplementary Figure 3).

****Referees cross-commenting****

I also agree that 4sU labeling may require too much work with a questionable result.

We are grateful to the reviewer for this comment, which helped us perform this reviewing in a timely manner.

Reviewer #3 (Significance (Required)):

Strengths:

1. Novelty of the described genomic arrangement of snoRNA/ncRNA genes and their processing in a sequential and regulated manner.
2. Potential conservation of this pathways across eukaryotic organisms.
3. Well designed and performed experiments followed by proper conclusions.

Limitations:

1. Insufficient evidence to support generalization of the study results.
2. Moderate overall impact of the study

Advance:

This research can be placed within publications describing specific processing pathways for various non-coding RNAs, including for example unusual chimeric species such as sno-lncRNAs. In this context, the presented results do advance the knowledge in the field by providing mechanistic evidence for a tightly controlled and coordinated maturation of selected ncRNAs.

Audience:

Basic research and specialized. The interest in this research will rather be limited to a specific field.

Dear Dr. Yague-Sanz,

Thank you for the submission of your revised manuscript. We have now received the enclosed reports from all referees and I am happy to say that all support its publication now. The referees still have a few more minor suggestions that I would like you to incorporate before we can proceed with the official acceptance of your manuscript.

A few editorial requests will also need to be addressed:

- Your ms has 4 main figures but the Results & Discussion sections are not combined. Please combine both sections to publish your paper as a short report.

- The ms needs to be uploaded as a Word file without figures and one column format. All main and EV figures need to be uploaded as individual production quality figure files. The suppl figures need to be uploaded as EV figures and all callouts need to be corrected. The nomenclature should be Figure EV1, etc. in all places (source files, titles in eJP, legends, ms callouts). You can find more info about our figure types in our guide to authors online.

- Please add up to 5 keywords to the ms file.

- Please correct the conflict of interest subheading to "Disclosure and Competing Interests Statement"

- The author credits need to be removed from the ms file. All credits need to be entered during online ms submission.

- The REFERENCE format needs to be alphabetical, not numerical; et al needs to be used after 10 author names

- You need to submit with your final ms a completed author checklist, which you can download from our author guidelines . The completed author checklist will also be part of the transparent peer-review process file.

- The grant number of the one funder, 40024367, is missing in the ms file, please add.

- The callout for Figure 1E is missing; Figure 3E is called out but the panel label is missing in the figure, Figure 4D is called out but the panel label is missing in the figure; we are not sure if Suppl. Figures 2 and 3 have been called out. Please correct.

- 3 suppl. tables are uploaded; Suppl. Table 1 is a dataset so needs to be updated to "Dataset EV1" and uploaded as such; Suppl. tables 2 and 3 can be updated to "Table EV1" and EV2; the nomenclature of all these needs to be updated in the table and the ms files.

- The Methods section needs to include a separate file called Reagents and Tools Table (listing key reagents, experimental models, software and relevant equipment and including their sources and relevant identifiers) and a Methods and Protocols section in which the methods are described using a step-by-step protocol format with bullet points, to facilitate the adoption of the methodologies across labs. More information on how to adhere to this format as well as downloadable templates (.docx) for the Reagents and Tools Table can be found in our author guidelines: <
<https://www.embopress.org/page/journal/14693178/authorguide#manuscriptpreparation>>.

- Discussion & Conclusions should be renamed to Discussion

- The manuscript sections should be in the following order: Title page - Abstract & Keywords - Introduction - Results - Discussion - Methods - Data Availability - Acknowledgments - Disclosure Statement & Competing Interests - References - Figure Legends - (Main Tables with legends if applicable) - Expanded View Figure Legends.

Figure Legends - Comments

- Please note that information related to n is missing in the legend of figure S2 A, please add.

- Please note that n=2 in figure S3 B. No statistics should be calculated if n=2. Please either repeat the experiment one more time or remove the error bars and p-values.

- Please note that the error bars are not defined in the legends of figures 2C, 3C, please add missing info.

- Please note that the exact p values are not provided in the legends of figures 2C, 3C, S2 A, S3B, please provide exact p-values as reasonable.

- Please add the specific URLs for GSE167041, SRR12004691, PRJEB83877, GSE12319, GSM7027460, SRR1995152 and GSE93651 datasets provided in the data availability section.

EMBO press papers are accompanied online by A) a short (1-2 sentences) summary of the findings and their significance, B) 2-3 bullet points highlighting key results and C) a synopsis image that is exactly 550 pixels wide and 200-600 pixels high (the height is variable). The synopsis image should provide a sketch of the major findings, like a graphical abstract. Please note that text needs to be readable at the final size. Please send us this information along with the final manuscript.

Referee #1:

The authors addressed all the issues. I don't have any further comments and recommend the manuscript for publication.

Referee #2:

I am satisfied with most of the answers and comments, as well as by additional analyses regarding conservation of snoRNA pairs processing pathway across eukaryotic organisms, that were provided by the authors. I still have one concern that should be addressed. First the sentence "The cleavage sites are located at the apical loop and flanking the stem, displaying a staggered symmetry characteristic of RNase III activity" on Page 7 is not accurate and should be rephrased. It suggests that also cleavages in the loop are performed by RNase III, which is a dsRNA-specific endonuclease. Also, the final conclusion "these analysis () ... are consistent with DROSHA, an RNase III enzyme, being responsible for this processing step" is somehow premature, because to my knowledge Drosha has not been reported to be involved in snoRNA biogenesis. There are known cases of miRNAs or other sRNAs produced from snoRNAs by DGCR8 independently of Drosha or RNA fragments generated from scaRNA by Drosha/DGCR8 and finally snoRNA degradation by the exosome recruited by DGCR8, but here DGCR8 or Drosha act on mature snoRNAs. Drosha contribution to snoRNA biogenesis is not completely unlikely, but if authors are aware of such cases, these papers should be referenced to. Otherwise, this suggestion should be treated with more caution. I also have one comment regarding the authors' reluctance to provide a northern quantification of the data, which is usually not a controversial point. Especially that quantification is now provided for the northern on Figure 2B, but still not for the northern on Figure 1B. Even if it is semi-quantitative, it is still more accurate than visual inspection. It seems that accumulation of U14 in Pac1-ts is more than slight. I agree that this effect can be due to characteristics of the ts strain, and it is not really relevant to the message of the MS, but I think that providing northern calculation is generally advisable for the sake of principle.

Minor corrections:

Page 7, "these analysis" it should be changed to "these analyses"

Supplementary Figure 2D, should be "rapamycin" instead of "ramapydin".

I recommend that the text and figures are more carefully scanned for spelling mistakes.

Referee #3:

I think the manuscript is significantly improved, although I would have organized it differently. I think the new data in supplemental figure 4D goes a long way to addressing my main comments 1 and 2 and the new version of supplemental figure S4A addresses my major comment 1. Thus, the authors have addressed all of my major comments. Having said that, I think supplemental figure 4D should replace main figure 4.

New minor comments are that:

I think "degradation should be replaced with "processing" in the second to last sentence of the abstract and again in the first sentence of the last paragraph of the results section.

Near the end of the second paragraph of the results section, the authors refer to figure 1C but I think they mean 1D.

"atypical" should be changed to "typical" in the phrase "the atypical 2nt- overhang signature of RNase III cleavage " on page 3.

REVISION EMBO REPORTS

Manuscript number: EMBOR-2025-62023V1-T

Corresponding author(s): Carlo, Yague-Sanz

1. General Statements [optional]

We thank the three reviewers and the editor for their constructive feedback, which, again, have substantially contributed to improving the quality of our revised manuscript. We believe that we have thoroughly addressed all the points raised.

To avoid confusion in the point-by-point response, we used a **black font** for the reviewer's comment and a **blue font** for our answer.

Referee #1:

The authors addressed all the issues. I don't have any further comments and recommend the manuscript for publication.

Referee #2:

I am satisfied with most of the answers and comments, as well as by additional analyses regarding conservation of snoRNA pairs processing pathway across eukaryotic organisms, that were provided by the authors. I still have one concern that should be addressed. First the sentence "The cleavage sites are located at the apical loop and flanking the stem, displaying a staggered symmetry characteristic of RNase III activity" on Page 7 is not accurate and should be rephrased. It suggests that also cleavages in the loop are performed by RNase III, which is a dsRNA-specific endonuclease.

We thank the reviewer for pointing this out. We have revised the sentence to clarify that the staggered symmetry pertains only to the cleavage sites flanking the stem. We also removed the phrase "*characteristic of RNase III activity*". The updated sentence now reads: "*The cleavage sites are located at the apical loop and flanking the stem, where they display a staggered symmetry.*"

Also, the final conclusion "these analysis () ... are consistent with DROSHA, an RNase III enzyme, being responsible for this processing step" is somehow premature, because to my knowledge Drosha has not been reported to be involved in snoRNA biogenesis. There are known cases of miRNAs or other sRNAs produced from snoRNAs by DGCR8 independently of Drosha or RNA

REVISION EMBO REPORTS

fragments generated from scaRNA by Drosha/DGCR8 and finally snoRNA degradation by the exosome recruited by DGCR8, but here DGCR8 or Drosha act on mature snoRNAs. Drosha contribution to snoRNA biogenesis is not completely unlikely, but if authors are aware of such cases, these papers should be referenced to. Otherwise, this suggestion should be treated with more caution.

We agree that we lack direct evidence for the involvement of DROSHA and have rephrased the sentence to more accurately reflect this uncertainty: *"These analyses [...] raise the possibility that the RNase III enzyme DROSHA is responsible for this processing step."*

I also have one comment regarding the authors' reluctance to provide a northern quantification of the data, which is usually not a controversial point. Especially that quantification is now provided for the northern on Figure 2B, but still not for the northern on Figure 1B. Even if it is semi-quantitative, it is still more accurate than visual inspection. It seems that accumulation of U14 in Pac1-ts is more than slight. I agree that this effect can be due to characteristics of the ts strain, and it is not really relevant to the message of the MS, but I think that providing northern calculation is generally advisable for the sake of principle.

We appreciate the reviewer's commitment to this principle and generally agree with it. However, quantifying each of the five mamRNA/snoU14 isoforms across the three main northern blot figures (Figures 1B, 2B, and 3B), as we did for snoU14 in Figure 2B (quantified in panel EV2A), would result in 14 additional panels. While it may be possible to condense this information, we believe that such extensive quantification is not justified in this context.

Moreover, we are not convinced that northern blot quantification is always more accurate than visual inspection. In our experience, the results can vary considerably depending on how regions of interest are defined and how background subtraction is performed. Instead, and as stated in our previous response to Reviewer 3, we prefer to support our main conclusions using orthogonal approaches, such as RT-qPCR or RNA-seq, which offer more robust and quantitative assessments of RNA accumulation.

That being said, we agree with the reviewer that the accumulation of U14 in the Pac1-ts mutant should not be described as "slight." While we originally meant this in relative terms compared to the accumulation of the precursor, it is not slight in absolute terms. We have revised the relevant section to reflect this more accurately.

Minor corrections:

Page 7, "these analysis" it should be changed to "these analyses"

Supplementary Figure 2D, should be "rapamycin" instead of "ramapydin".

I recommend that the text and figures are more carefully scanned for spelling mistakes.

We thank the reviewer for pointing out these spelling errors. We have corrected the text and figures accordingly.

REVISION EMBO REPORTS

Referee #3:

I think the manuscript is significantly improved, although I would have organized it differently. I think the new data in supplemental figure 4D goes a long way to addressing my main comments 1 and 2 and the new version of supplemental figure S4A addresses my major comment 1. Thus, the authors have addressed all of my major comments. Having said that, I think supplemental figure 4D should replace main figure 4.

We thank the reviewer for this excellent suggestion. We have moved the previous Figures 4B–C to Figures EV4D–E and relocated the previous Figure S4D to Figure 4B. These changes have improved the flow and presentation of the manuscript.

New minor comments are that:

I think "degradation should be replaced with "processing" in the second to last sentence of the abstract and again in the first sentence of the last paragraph of the results section.

Near the end of the second paragraph of the results section, the authors refer to figure 1C but I think they mean 1D.

"atypical" should be changed to "typical" in the phrase "the atypical 2nt- overhang signature of RNase III cleavage " on page 3.

We thank the reviewer for pointing out these errors. We have corrected the text accordingly.

Dr. Carlo Yague-Sanz
University of Namur
URPHYM-GEMO, NARILIS
Namur
Belgium

Dear Dr. Yague-Sanz,

I am very pleased to accept your manuscript for publication in the next available issue of EMBO reports. Thank you for your contribution to our journal.

Yours sincerely,
